# FlexRound: Learnable Rounding by Element-wise Division for Post-Training Quantization

## Abstract

Post-training Quantization (PTQ) has been gaining popularity for the deployment of deep neural networks on resource-limited devices since unlike quantization-aware training, neither a full training dataset nor end-to-end training is required at all. As PTQ schemes based on reconstructing each layer or block output turn out to be effective to enhance quantized model performance, recent works have developed algorithms to devise and learn a new weight-rounding scheme so as to better reconstruct each layer or block output. We notice that, however, such new rounding schemes are established on element-wise addition. In this work, we propose a simple yet effective new rounding mechanism for post-training weight quantization, coined *FlexRound*, via element-wise division to learn not only a common quantization grid size but also a different scale for each pre-trained weight. Thanks to the reciprocal rule of derivatives induced by element-wise division, FlexRound is inherently able to exploit the importance of a pre-trained weight when updating its corresponding scale, and thus, flexibly quantize a pre-trained weight depending on its own importance. We empirically validate the efficacy of FlexRound on a wide range of models and tasks. To the best of our knowledge, our work is the first to carry out comprehensive experiments on not only image classification and natural language understanding but natural language generation in the *per-tensor* uniform PTQ setting. Our code will be open-sourced soon.

## 1 Introduction

Recent years have witnessed the unprecedented success of deep neural networks in a wide variety of domains including computer vision, natural language processing, automatic speech recognition, and so on. Although state-of-the-art deep neural networks surpass human-level performance, these neural networks cannot help requiring more and more computation cost and memory usage as networks become deeper and wider. In order to reduce the model size and accelerate inference operations, many researchers have attempted diverse compression techniques such as network quantization (Courbariaux et al., 2016) and network pruning (Han et al., 2016). In this paper, we concentrate on network quantization due to the advantage that INT4 or INT8 quantization allows us to accelerate quantized neural networks using off-the-shelf accelerators such as the NVIDIA A100 Tensor Core GPU (Wu et al., 2020) or ARM Cortex MCUs (Kim et al., 2021).

Network quantization techniques can be generally divided into two categories: quantization-aware training (QAT) and post-training quantization (PTQ). When quantizing neural networks via QAT (Jung et al., 2019; Jain et al., 2019; Zhao et al., 2020; Esser et al., 2020; Lee et al., 2021), the performance gap between a full-precision neural network and its quantized counterpart can be marginal. Yet, QAT requires end-to-end retraining or fine-tuning on a full training dataset, which often causes an enormous amount of time and resources to obtain a quantized neural network with competitive performance. Furthermore, a whole training dataset may not be available due to data privacy issues or demands to utilize legacy models. Such drawbacks of QAT are the reasons why

researchers recently pay more attention to PTQ (Zhao et al., 2019; Wang et al., 2020; Nahshan et al., 2021) that needs neither a full training dataset nor end-to-end learning at all.

PTQ had been initially performed via rounding-to-nearest scheme by minimizing the quantization error in the parameter space. Unfortunately, this approach suffers from severe performance degradation. Since it is reported that the loss degradation resulting from quantization can be approximated as the second-order error in Taylor Expansion by viewing quantized weights as perturbed weights, Nagel et al. (2020) and Li et al. (2021) substantiate that reconstructing each output of layer or block is equivalent to minimizing the approximation of loss degradation resulting from quantization under some assumptions. Accordingly, recent works (Nagel et al., 2020; Li et al., 2021; Hubara et al., 2021; Wei et al., 2022) have suggested to reconstruct each output of layer or block by devising and learning a new weight-rounding scheme, deviating from rounding-to-nearest, as an effort to preserve the performance of a full-precision model. However, all those new rounding schemes designed in existing studies either round or quantize pre-trained weights adaptively via element-wise addition.

Changing the perspective of a new rounding policy from element-wise addition to element-wise division, we propose a simple yet effective post-training weight quantization method called FlexRound, which flexibly quantizes pre-trained weights by learning how much each pre-trained weight should be divided by. Interestingly, thanks to the reciprocal rule of derivatives induced by element-wise division, FlexRound can inherently leverage pre-trained weights when updating an individual scale for every pre-trained weight. Specifically, we corroborate that a relatively wider range of discrete values needs to be explored when quantizing pre-trained weights of large magnitude. The rationale behind such an approach is that the magnitude of weight can be considered as its importance. Given that it is crucial to retain the knowledge of important weights even after quantization so as to maintain the performance of a pre-trained model, the constraints associated with quantizing weights of large absolute value should be relaxed compared to those of small absolute value (i.e., those important weights can be quantized to one of not only its two nearest discrete values but also discrete ones far from it). Accordingly, FlexRound quantizes pre-trained weights flexibly depending on each their own importance, thereby leading to better performance.

Our contributions are threefold:

- We propose FlexRound as a new rounding scheme for post-training weight quantization based on the principle of element-wise division to enable learning separate scales for all pre-trained weights as well as a common quantization grid size across a group (e.g., a channel or a layer).

- We demonstrate that such a new rounding scheme via element-wise division takes into consideration the importance of pre-trained weights when updating their corresponding scales so that FlexRound can quantize pre-trained weights of large magnitude (i.e., important pre-trained weights) more flexibly.

- To the best of our knowledge, we are the first to conduct extensive experiments in the form of *per-tensor* uniform PTQ reconstruction on natural language generation as well as image classification and natural language understanding. We verify the effectiveness of FlexRound using numerous models such as ResNet, MobileNetV2, BERT, GPT-Neo, and OPT.

## 2 RELATED WORK

Recently, many researchers have attempted to quantize a wide range of models for various tasks such as vision and language understanding/generation without any (re)training. OCS (Zhao et al., 2019) replicates channels entailing outliers, and then, halves outliers of those channels. Unfortunately, even though OCS explicitly addresses outliers, it still suffers from severe accuracy degradation when both weights and activations are quantized into low-bit. As an alternative solution, Wang et al. (2020) proposed Bit-Split that splits an integer into several bits and optimizes them separately. Although Wang et al. (2020) showed that the performance of Bit-Split is close to that of a full-precision model in the low-bit setting, Bit-Split may not be effective for certain architectures including MobileNetV2.

To overcome the limitations discussed above, Nagel et al. (2020) and Hubara et al. (2021) minimize the mean squared error (in a layer-by-layer fashion) between the full-precision layer's output and its quantized layer's output by inventing and learning a new weight-rounding mechanism dubbed as AdaRound and AdaQuant, respectively. As such a layer-wise reconstruction error minimization opens the door to 4-bit PTQ regime, Li et al. (2021) proposed block-wise reconstruction, titled BRECQ, to consider cross-layer dependency along with the possibility of fully quantizing MobileNetV2 into 4-bit. In addition to block-wise reconstruction, Wei et al. (2022) proposed QDrop that drops the quantization of activations at random during reconstruction to induce activation quantization to be synchronized with weight quantization. Both BRECQ and QDrop, however, are based on AdaRound, which cannot learn a quantization grid size while quantizing weights allows for rounding either up or down only at most. AdaQuant quantizes weights adaptively. AdaQuant, however, does not consider the magnitude of weights for quantization that turns out to be important as we discuss later.

As another line of post-training quantization (PTQ) research, some PTQ techniques are specialized in quantizing language models such as BERT and GPT-like models. Bondarenko et al. (2021) first applied PTQ to BERT by introducing per-embedding-group activation quantization scheme to deal with highly dynamic activation ranges. Bai et al. (2021) studied the PTQ reconstruction in parallel for BERT. Yao et al. (2022) proposed ZeroQuant that quantizes BERT and GPT-3 in group-wise weight quantization manner driven by token-wise activation quantization via layer-by-layer knowledge distillation. Dettmers et al. (2022) quantizes large language models like OPT with vector-wise weight quantization and mixed-precision decomposition with FP16 activation. All those methods do not consider per-tensor weight quantization which can enable integer matrix-to-matrix multiplication API/function calls (Migacz, 2017).

Most of the aforementioned PTQ studies are targeted to either vision models or language models only, but not to both. Most experimental results in the above PTQ works are conducted via channel-wise/group-wise/vector-wise weight quantization at the expense of reduced parallelism. To the best of our knowledge, our work is the first to carry out extensive experiments on diverse tasks ranging from image classification to natural language generation assuming a per-tensor uniform PTQ setting.

## 3 METHODOLOGY

In this section, we first present the notations used in the paper, describe the concept and design of FlexRound for per-tensor uniform post-training quantization (PTQ) reconstruction, and then, scrutinize how FlexRound can leverage the importance of a pre-trained weight.

### 3.1 PRELIMINARIES

**Notations.** A scalar, a vector, and a matrix (or a tensor) are expressed as a non-bold letter, a small bold letter and a capital bold letter (e.g. $s$, $\boldsymbol{s}$ and $\boldsymbol{S}$) respectively. $\widehat{\boldsymbol{W}}$ indicates the quantized counterpart of $\boldsymbol{W}$. The input to a convolutional or fully-connected layer is denoted as $\boldsymbol{X}$ if all previous layers are intact or as $\widetilde{\boldsymbol{X}}$ if all previous layers are quantized. The $(i, j)$ element of a matrix $\boldsymbol{W}$ is represented as $W_{(i,j)}$. We let $\odot$ and $/$ indicate element-wise product and element-wise division, respectively, similar to the broadcasting process in Python Numpy. $\lfloor \cdot \rceil$ and $\lfloor \cdot \rfloor$ express the rounding function and the floor function. $\| \cdot \|_F$ represents the Frobenius norm.

**PTQ Background.** The conventional uniform PTQ approach is to quantize pre-trained weights $\boldsymbol{W}$ to be $\widehat{\boldsymbol{W}} = s_1 \left\lfloor \frac{\boldsymbol{W}}{s_1} \right\rceil$ via rounding-to-nearest and to minimize $\|\boldsymbol{W} - \widehat{\boldsymbol{W}}\|_F^2$ with respect to the quantization grid size $s_1$, but the minimization of quantization error in the parameter space is not equivalent to that of the final task loss. On the grounds that Li et al. (2021) proves that the loss degradation resulting from quantization can be approximated as the quadratic form of the network output and its Hessian matrix, several existing studies have strove to minimize $\|\boldsymbol{W}\boldsymbol{X} - \widehat{\boldsymbol{W}}\widetilde{\boldsymbol{X}}\|_F^2$ layer-by-layer or block-by-block with respect to continuous variables $\boldsymbol{V}$ with only a small amount

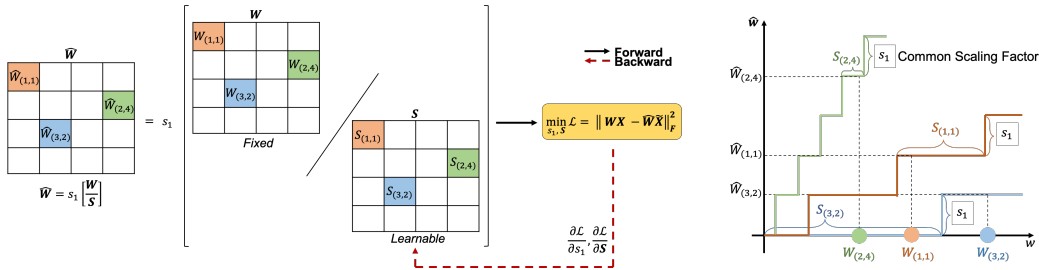

(a) A new rounding scheme via element-wise division. Both $s_1$ and $\boldsymbol{S}$ are updated toward minimizing the reconstruction error, $\mathcal{L}$.

(b) Rounding functions with learned parameters $s_1$ and $\boldsymbol{S}$ as shown in (a).

Figure 1: Illustration of FlexRound in the per-tensor uniform PTQ reconstruction. As seen in (b), FlexRound flexibly quantizes pre-trained weights by observing $W_{(2,4)} < W_{(3,2)}$ but $\widehat{W}_{(2,4)} > \widehat{W}_{(3,2)}$.

of data, where $\widehat{\boldsymbol{W}}$ is either $s_1(\lfloor \frac{\boldsymbol{W}}{s_1} \rfloor + h(\boldsymbol{V}))$ with a certain function $h(\cdot)$ (Nagel et al., 2020) or $s_1 \lfloor \frac{\boldsymbol{W}+\boldsymbol{V}}{s_1} \rceil$ (Hubara et al., 2021). However, all these aforementioned rounding mechanisms are founded on element-wise addition.

## 3.2 FlexRound

Unlike prior works based on element-wise addition, we exploit element-wise division for quantizing pre-trained weights. We can formulate our proposed weight-rounding scheme via element-wise division as follows:

$$\widehat{\boldsymbol{W}} = s_1 \left\lfloor \frac{\boldsymbol{W}}{\boldsymbol{S}} \right\rceil, \tag{1}$$

where the shape of $\boldsymbol{S}$ is equal to that of $\boldsymbol{W}$ while all entries of $\boldsymbol{S}$ as well as the quantization grid size $s_1$ are positive and learnable. Similarly to preceding studies, both $s_1$ and $\boldsymbol{S}$ are updated as an attempt to minimize $\|\boldsymbol{W}\boldsymbol{X} - \widehat{\boldsymbol{W}}\widetilde{\boldsymbol{X}}\|_F^2$.

Eq. 1 implies that the basic formula of FlexRound supports per-tensor uniform PTQ. Notice that although FlexRound can adopt a per-channel weight quantization scheme simply by replacing a scalar $s_1$ with a vector $\boldsymbol{s}_1$, since we show later that per-tensor uniform PTQ (using FlexRound) is enough to provide the accuracy of a full-precision model, we set a single quantization grid size $s_1$ for each layer (Per-tensor quantization schemes might enable integer matrix-to-matrix multiplication API/function calls that can facilitate efficient inference of quantized models. (Migacz, 2017)). From now on, thus, we study only the per-tensor uniform PTQ reconstruction. The overall procedure of FlexRound is described in Figure 1.

Now let us discuss how to design $\boldsymbol{S}$. Let $\boldsymbol{W} \in \mathbb{R}^{C_{out} \times C_{in}}$ in the case of a fully-connected layer and $\boldsymbol{W} \in \mathbb{R}^{C_{out} \times C_{in} \times H \times W}$ in the case of a convolutional layer. We first start formulating $\boldsymbol{S}$ as $\boldsymbol{S} = s_1 \odot \boldsymbol{S}_2$ where $\boldsymbol{S}_2 \in \mathbb{R}_{>0}^{C_{out} \times C_{in}}$ in the case of a fully-connected layer and $\boldsymbol{S}_2 \in \mathbb{R}_{>0}^{C_{out} \times C_{in} \times H \times W}$ in the case of a convolutional layer while all elements

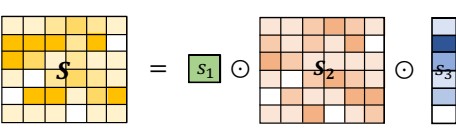

Figure 2: Formation of $\boldsymbol{S}$ for a linear layer.

of $\boldsymbol{S}_2$ are learnable. Then, motivated by a wide acknowledgement that the statistics of output channels can vary greatly (Nagel et al., 2019; Lou et al., 2020), we account for the variation of output channel's statistics by complementing $\boldsymbol{S}$ with an additional learnable tensor $\boldsymbol{s}_3$, where $\boldsymbol{s}_3 \in \mathbb{R}_{>0}^{C_{out} \times 1}$ in the case of a fully-connected layer and $\boldsymbol{s}_3 \in \mathbb{R}_{>0}^{C_{out} \times 1 \times 1 \times 1}$ in the case of a convolutional layer. For a convolutional layer, $\boldsymbol{S}$ is additionally complemented by another learnable tensor $\boldsymbol{s}_4$, where $\boldsymbol{s}_4 \in \mathbb{R}_{>0}^{1 \times C_{in} \times 1 \times 1}$. Consequently, $\boldsymbol{S}$ is formulated as $s_1 \odot \boldsymbol{S}_2 \odot \boldsymbol{s}_3$ for a fully-connected layer as displayed in Figure 2 and $s_1 \odot \boldsymbol{S}_2 \odot \boldsymbol{s}_3 \odot \boldsymbol{s}_4$ for a convolutional layer.

Accordingly, quantization process for FlexRound can be expressed as

$$\widehat{\boldsymbol{W}} = \begin{cases} s_1 \left\lfloor \dfrac{\boldsymbol{W}}{s_1 \odot \boldsymbol{S}_2 \odot \boldsymbol{s}_3} \right\rceil & \text{if } \boldsymbol{W} \text{ is a fully-connected layer} \\ s_1 \left\lfloor \dfrac{\boldsymbol{W}}{s_1 \odot \boldsymbol{S}_2 \odot \boldsymbol{s}_3 \odot \boldsymbol{s}_4} \right\rceil & \text{if } \boldsymbol{W} \text{ is a convolutional layer} \end{cases} \quad (2)$$

where all entries of $\boldsymbol{S}_2$, $\boldsymbol{s}_3$, and $\boldsymbol{s}_4$ are initialized to be ones in order to enable learning $\boldsymbol{S}_2$, $\boldsymbol{s}_3$, and $\boldsymbol{s}_4$ from rounding-to-nearest, $s_1 \left\lfloor \frac{\boldsymbol{W}}{s_1} \right\rceil$. $s_1$, $\boldsymbol{S}_2$, $\boldsymbol{s}_3$, and $\boldsymbol{s}_4$ are updated to minimize $\|\boldsymbol{W}\boldsymbol{X} - \widehat{\boldsymbol{W}}\widetilde{\boldsymbol{X}}\|_F^2$ subject to the constraint that all elements of $s_1$, $\boldsymbol{S}_2$, $\boldsymbol{s}_3$, and $\boldsymbol{s}_4$ are positive.

Since $s_1$, $\boldsymbol{S}_2$, $\boldsymbol{s}_3$, and $\boldsymbol{s}_4$ are all learnable and FlexRound does not need any explicit regularization terms, no additional hyper-parameter is necessary, and thus, FlexRound would be convenient for practitioners. Moreover, as all entries of $s_1$, $\boldsymbol{S}_2$, $\boldsymbol{s}_3$, and $\boldsymbol{s}_4$ are positive and FlexRound is based on element-wise division, FlexRound encourages $\widehat{\boldsymbol{W}}$ to employ the same sign as $\boldsymbol{W}$. Hence, FlexRound prevents extreme changes of weights through quantization process unlike some element-wise addition rounding scheme such as AdaQuant (Hubara et al., 2021).

## 4  EXPERIMENTS

In this section, we present experimental results for benchmark datasets and network models in computer vision and natural language processing tasks. We first empirically confirm that additional tensors $\boldsymbol{s}_3$ and $\boldsymbol{s}_4$ introduced in Section 3.2 implement distinct contributions in the per-tensor uniform post-training quantization (PTQ) setting. Then, we compare the performance of FlexRound with that of some state-of-the-art PTQ approaches in the following cases: image classification on the ImageNet (Russakovsky et al., 2015) dataset with the ResNet (He et al., 2016) and MobileNetV2 (Sandler et al., 2018) architectures (Section 4.3), natural language understanding (NLU) on the GLUE (Wang et al., 2018) benchmark with the BERT (Devlin et al., 2018) and GPT-Neo (Black et al., 2021) architectures (Section 4.4), and natural language generation (NLG) on WikiText2 (Merity et al., 2016) and Penn Treebank (PTB) (Marcus et al., 1993) with the GPT-Neo and OPT (Zhang et al., 2022) architectures (Section 4.4). For brevity, we let "B + X" and "Q + X" indicate that a certain rounding scheme 'X' is performed in the experimental setup described in BRECQ (Li et al., 2021) or QDrop (Wei et al., 2022), respectively (an experimental setup includes the definition of a block unit for reconstruction error minimization or how much the probability of dropping the quantization of activations is). As introduced in BRECQ and QDrop, we also utilize the LSQ technique (Esser et al., 2020) when updating an activation step size for activation quantization. Throughout our comprehensive experiments, we verify that FlexRound can achieve competitive performance with a full-precision model for the above tasks even in the *per-tensor* uniform PTQ reconstruction, which has not been introduced previously. All experimental results in this section are conducted by our own implementation based on open-source codes.

### 4.1  LEVERAGING THE IMPORTANCE OF A PRE-TRAINED WEIGHT

As we discussed previously, either element-wise addition or element-wise division is effective to produce a better rounding scheme than a rounding to the nearest scheme. In order to investigate the difference between element-wise addition and element-wise division, it would be instructive to analyze the gradient of the reconstruction error $\mathcal{L} = \|\boldsymbol{W}\boldsymbol{X} - \widehat{\boldsymbol{W}}\widetilde{\boldsymbol{X}}\|_F^2$ with respect to $\boldsymbol{S}'$ (where $\boldsymbol{S}'$ is $\boldsymbol{S}_2 \odot \boldsymbol{s}_3$ for a fully-connected layer and $\boldsymbol{S}_2 \odot \boldsymbol{s}_3 \odot \boldsymbol{s}_4$ for a convolutional layer). Through analysis, unlike element-wise addition, we show that element-wise division enables $\frac{\partial \mathcal{L}}{\partial \boldsymbol{S}'}$ to leverage the importance of pre-trained weights $\boldsymbol{W}$, as follows[1]:

Using the straight-through estimator (Bengio et al., 2013), for every $i$ and $j$, $\left| \frac{\partial \mathcal{L}}{\partial S'_{(i,j)}} \right|$ is directly proportional to $\left| W_{(i,j)} \frac{\partial \mathcal{L}}{\partial \widehat{W}_{(i,j)}} \right|$, which implies that $S'_{(i,j)}$ is (partially) affected by $W_{(i,j)}$. As a result,

---

[1]For simplicity, we take into account the case of a fully-connected layer.

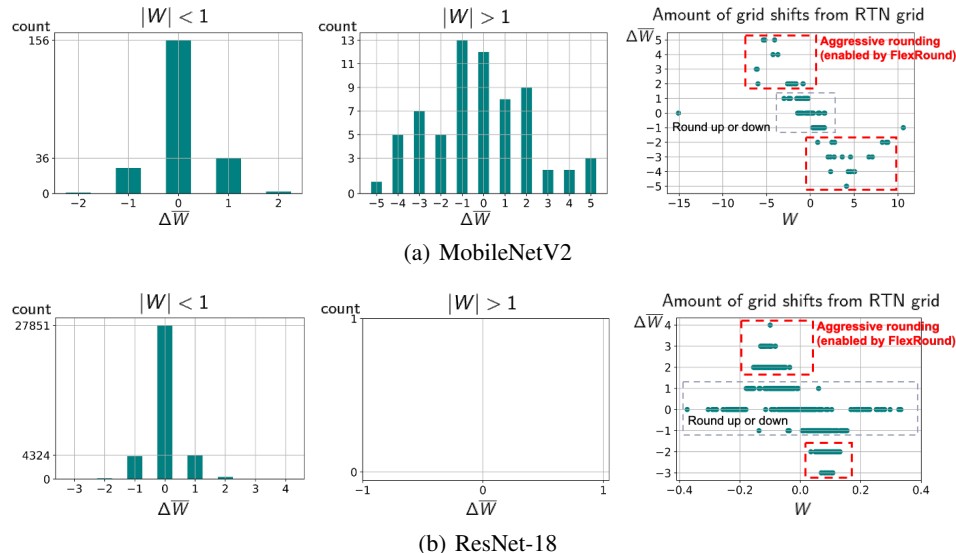

(a) MobileNetV2

(b) ResNet-18

Figure 3: Weight updates through FlexRound of the first convolutional layer in the first block of (a) MobileNetV2 and (b) ResNet-18, after quantizing pre-trained weights into 4-bit (by FlexRound) while activations are kept in full-precision.

$\overline{W}_{(i,j)} = \left\lfloor \frac{W_{(i,j)}}{s_1 \odot S'_{(i,j)}} \right\rceil$ can also be updated and influenced by $W_{(i,j)}$ as well. In other words, as the magnitude of a pre-trained weight $W_{(i,j)}$ is larger, the chance of $\overline{W}_{(i,j)}$ receiving a larger update becomes higher during the PTQ reconstruction. In light of the fact that the magnitude of a weight can be regarded as a metric to measure importance during compressing a neural network (Han et al., 2015; Zhu & Gupta, 2017), if the goal is to enhance model accuracy after quantization, it would be reasonable to have less important (that is, smaller magnitude) weights rounded either up or down only while allowing more important (i.e., exhibiting larger magnitude) weights to be quantized to one of the two closest quantization grids or more.

Figure 3 presents the amount of weight updates through FlexRound for MobileNetV2 and ResNet-18. On the left side and the center side of Figure 3, histograms describe the change of $\overline{W}_{(i,j)}$ grouped for small pre-trained weights ($|W| < 1$, left) and large pre-trained weights ($|W| > 1$, center). On the right side, scatter plots show the amount of grid shifts from the grids obtainable by the rounding-to-nearest (RTN) scheme. We note that MobileNetV2 and ResNet-18 are quantized distinctively due to FlexRound. For example, in the case of MobileNetV2 as illustrated in Figure 3(a), the change of $\overline{W}_{(i,j)}$ attained by minimizing $\mathcal{L}$ is more aggressive (i.e., rounding can be deviated by more than one-step up or one-step down) when the absolute value of $W_{(i,j)}$ is larger than one, which means that FlexRound more flexibly quantizes pre-trained weights of large magnitude as illustrated in red dotted squares in Figure 3(a). The amount of aggressively rounded weights in the first convolutional layer of the first block of MobileNetV2 is around $12.8\%$ of the total. For ResNet-18, however, there are no pre-trained weights whose magnitudes are larger than one. Thus, most pre-trained weights are rounded either up or down as shown in Figure 3(b) (e.g., only about $1.5\%$ weights are rounded aggressively in the first convolutional layer of the first block of ResNet-18). Different rounding results by FlexRound, AdaRound, and AdaQuant are visually compared in Appendix A.

## 4.2 ABLATION STUDY

To justify the introduction of $s_3$ and $s_4$ on FlexRound in the per-tensor uniform PTQ setting, we investigate the impact of $s_3$ and $s_4$ on the performance of FlexRound using the ImageNet dataset with pre-trained weights quantized into 2-bit (activations are not quantized). As shown in the last two rows in Table 1, the presence of $s_3$ and $s_4$ enhances the accuracy for all models. Interestingly, FlexRound outperforms both AdaQuant and AdaRound even without $s_3$ and $s_4$, which would support

Table 1: Top-1/Top-5 accuracy (%) on ImageNet by ResNet-18, ResNet-50, and MobileNetV2 with only weights quantized into 2-bit. "B + X" denotes the implementation of X in the setting of BRECQ. We employ pre-trained models available from the official PyTorch repository.

| Method | ResNet-18 | ResNet-50 | MobileNetV2 |
|---|---|---|---|
| B + AdaQuant | 1.13/4.10 | 0.12/0.60 | 0.10/0.50 |
| B + AdaRound | 63.01/85.20 | 68.31/88.98 | 33.10/60.58 |
| B + FlexRound without $s_3$, $s_4$ | 63.19/85.08 | 70.00/89.82 | 34.75/62.51 |
| B + FlexRound with $s_3$, $s_4$ | **63.73/85.41** | **70.57/90.07** | **38.09/64.90** |

Table 2: Top-1/Top-5 accuracy (%) for ResNet-18, ResNet-50, and MobileNetV2 on ImageNet when only weights are quantized. "B + X" expresses the implementation of X in the BRECQ's setting. We employ pre-trained models available from the BRECQ github repository

| Method | # Bits (W./A.) | ResNet-18 | ResNet-50 | MobileNetV2 |
|---|---|---|---|---|
| Full-precision | 32/32 | 71.00/89.97 | 76.63/93.04 | 72.62/90.67 |
| B + AdaQuant | 4/32 | 67.50/87.75 | 72.79/90.77 | 15.17/32.89 |
| B + AdaRound | 4/32 | 70.18/89.38 | 75.86/92.62 | 69.46/88.85 |
| B + FlexRound (Ours) | 4/32 | **70.28/89.44** | **75.95/92.68** | **70.82/89.67** |
| B + AdaQuant | 3/32 | 57.09/80.82 | 52.13/75.22 | 0.20/0.79 |
| B + AdaRound | 3/32 | **68.79/88.62** | 74.31/91.81 | 62.51/84.52 |
| B + FlexRound (Ours) | 3/32 | 68.65/88.54 | **74.38/91.81** | **66.87/87.56** |
| B + AdaQuant | 2/32 | 0.23/0.92 | 0.10/0.50 | 0.10/0.50 |
| B + AdaRound | 2/32 | 61.99/84.81 | 48.47/77.09 | 39.57/66.18 |
| B + FlexRound (Ours) | 2/32 | **62.57/84.84** | **63.67/85.72** | **46.04/72.48** |

our claim that a new rounding scheme, shifted from element-wise addition to element-wise division, is the key to improving quantization quality significantly.

### 4.3 RESNET-18, RESNET-50, AND MOBILENETV2 ON IMAGENET

In this subsection, we quantize ResNet-18, ResNet-50, and MobileNetV2 in the low-bit PTQ reconstruction with 1024 randomly sampled images. Linear symmetric per-tensor quantization format is assumed to quantize weights and/or activations. For FlexRound, the output of each layer or block is reconstructed during 5k iterations while all learnable parameters (i.e., $s_1$, $S_2$, $s_3$, and $s_4$) are updated by using one learning rate (e.g., 4e-4 for the ResNet models quantized by 3-bit or 4-bit, or 1e-3 for the ResNet models quantized by 2-bit and MobileNetv2). The first and last layers are quantized into 8-bit and the batch normalization layer is folded into convolution, as done in Li et al. (2021). Our experiments are performed based on full-precision pre-trained models available from the BRECQ (Li et al., 2021) github repository[2], and we report the median over five random trials.

Assuming the quantization of weights only, we compare FlexRound with AdaRound and AdaQuant that utilize the principle of element-wise addition to decide rounding operations. Table 2 shows that FlexRound consistently outperforms those two addition-based rounding policies. Note that the performance of AdaQuant is inferior to that of AdaRound in Table 2. Correspondingly, FlexRound would be compared to AdaRound only to save space hereafter. Table 3 provides model accuacy when AdaRound and FlexRound (to quantize both weights and activations) are associated with the settings of BRECQ or QDrop. In Table 3, it should be noted that FlexRound is particularly successful for MobileNetV2 incorporating weights of large magnitude, for the reason that we explained in Section 4.1. It is also interesting to see that even when both weights and activations of the ResNet

---

[2]https://github.com/yhhhli/BRECQ

Table 3: Top-1/Top-5 accuracy (%) for ResNet-18, ResNet-50, and MobileNetV2 on ImageNet when both weights and activations are quantized. "B + X" and "Q + Y" represent the implementation of X in the BRECQ's setting and that of Y in the QDrop's setting, respectively. We employ pre-trained models available from the BRECQ github repository.

| Method | # Bits (W./A.) | ResNet-18 | ResNet-50 | MobileNetV2 |
|---|---|---|---|---|
| Full-precision | 32/32 | 71.00/89.97 | 76.63/93.04 | 72.62/90.67 |
| B + AdaRound | 4/4 | 69.18/88.85 | 74.44/91.80 | 61.05/83.30 |
| B + FlexRound (Ours) | 4/4 | **69.32/88.83** | 74.56/91.87 | 63.74/85.01 |
| Q + AdaRound | 4/4 | 69.20/88.96 | 74.90/92.15 | 65.42/86.23 |
| Q + FlexRound (Ours) | 4/4 | 69.26/88.81 | **75.08/92.20** | **66.66/87.21** |
| B + AdaRound | 3/3 | 64.83/86.12 | 67.01/87.28 | 3.74/11.54 |
| B + FlexRound (Ours) | 3/3 | 64.99/85.93 | 68.29/87.89 | 25.43/48.28 |
| Q + AdaRound | 3/3 | **65.71/86.96** | 70.49/89.93 | 39.86/66.00 |
| Q + FlexRound (Ours) | 3/3 | 65.43/86.60 | **70.74/89.78** | **51.49/76.90** |

Table 4: Performance of $BERT_{Base}$, $BERT_{Large}$, on the GLUE benchmark. For evaluation metrics, matched and mismatched accuracies are reported for MNLI, F1 score and accuracy are reported for QQP, Mathews correlation is reported for CoLA, Pearson and Spearman correlations are reported for STS-B, and accuracy is reported for the others. "Q + X" indicates the implementation of X in the QDrop's setting.

| Dataset | Method | $BERT_{BASE}$ | $BERT_{LARGE}$ | $GPT\text{-}Neo_{125M}$ | $GPT\text{-}Neo_{1.3B}$ | $GPT\text{-}Neo_{2.7B}$ |
|---|---|---|---|---|---|---|
| MNLI | Full-precision | 84.49/85.20 | 86.05/85.98 | 79.11/79.63 | 85.12/86.04 | 86.36/87.02 |
| | Q+AdaRound | 83.69/84.61 | 85.75/85.86 | 72.67/74.11 | 84.90/85.82 | 86.33/86.75 |
| | Q+FlexRound (Ours) | **84.53/84.98** | **85.93/85.99** | **72.94/74.24** | **85.56/86.14** | **86.41/86.89** |
| QQP | Full-precision | 88.06/91.08 | 88.66/91.59 | 85.20/88.99 | 88.26/91.28 | 88.62/91.50 |
| | Q+AdaRound | 87.65/90.58 | 87.48/90.62 | 72.97/79.35 | 87.98/91.04 | 88.38/91.27 |
| | Q+FlexRound (Ours) | **87.81/90.83** | **88.38/91.31** | **73.75/80.65** | **88.27/91.18** | **88.60/91.39** |
| QNLI | Full-precision | 91.25 | 92.13 | 85.15 | 91.36 | 92.46 |
| | Q+AdaRound | 91.16 | **92.24** | **80.87** | 91.40 | 92.04 |
| | Q+FlexRound (Ours) | **91.16** | 92.04 | 80.52 | **91.54** | **92.50** |
| SST-2 | Full-precision | 93.00 | 92.78 | 89.91 | 93.35 | 94.50 |
| | Q+AdaRound | **92.66** | 93.00 | **84.75** | 92.55 | 93.81 |
| | Q+FlexRound (Ours) | 92.43 | **93.58** | 83.03 | **93.12** | **94.04** |
| CoLA | Full-precision | 58.55 | 63.57 | 37.83 | 57.42 | 58.88 |
| | Q+AdaRound | 56.79 | 54.30 | 20.15 | 58.93 | 57.14 |
| | Q+FlexRound (Ours) | **57.53** | **60.57** | **21.59** | **59.30** | **57.37** |
| STS-B | Full-precision | 88.52/88.20 | 88.98/88.89 | 79.87/80.12 | 88.94/88.90 | 89.75/89.82 |
| | Q+AdaRound | 88.00/87.53 | 86.87/86.69 | **68.55**/68.25 | **88.97/88.77** | 89.03/**88.91** |
| | Q+FlexRound (Ours) | **88.29/87.91** | **88.82/88.76** | 67.65/**68.34** | 88.82/88.58 | **89.06**/88.69 |
| MRPC | Full-precision | 85.05 | 85.54 | 80.15 | 85.05 | 87.99 |
| | Q+AdaRound | 81.62 | 82.35 | 75.25 | 84.80 | 85.78 |
| | Q+FlexRound (Ours) | **84.07** | **84.31** | **75.49** | **85.05** | **86.76** |
| RTE | Full-precision | 64.62 | 71.19 | 64.98 | 76.17 | 80.87 |
| | Q+AdaRound | 63.54 | 66.79 | 62.82 | 75.09 | 80.51 |
| | Q+FlexRound (Ours) | **64.62** | **68.95** | **62.82** | **76.17** | **81.23** |

models are quantized into 4-bit under the per-tensor uniform PTQ setting, the performance degradation (compared to a full-precision pre-trained model) is negligible (less than 1.5%) in Table 3.

## 4.4 LANGUAGE MODELS

All language models we consider in this paper are based on the structure of Transformers (Vaswani et al., 2017). To quantize Transformers into 8-bit, we apply linear asymmetric per-tensor quantization scheme for both weights and activations, while reconstruction (for PTQ) is considered for each

Table 5: Performance of GPT-Neo$_{125M}$, GPT-Neo$_{1.3B}$, GPT-Neo$_{2.7B}$, OPT$_{125M}$, OPT$_{1.3B}$ and OPT$_{2.7B}$ on the WikiText2 and PTB datasets. The perplexity (PPL) is employed as a performance metric. The lower PPL, the better. "Q + X" means the implementation of X in the QDrop's setting.

| Dataset | Method | GPT-Neo$_{125M}$ | GPT-Neo$_{1.3B}$ | GPT-Neo$_{2.7B}$ | OPT$_{125M}$ | OPT$_{1.3B}$ | OPT$_{2.7B}$ |
|---|---|---|---|---|---|---|---|
| WikiText2 | Full-precision | 31.54 | 15.40 | 13.35 | 56.08 | 29.76 | 26.13 |
| | Q+AdaRound | 35.60 | 15.75 | 13.95 | 226.48 | 40.40 | 47.48 |
| | Q+FlexRound (Ours) | **33.44** | **15.68** | **13.80** | **66.07** | **40.01** | **40.38** |
| PTB | Full-precision | 64.63 | 31.51 | 27.22 | 129.90 | 76.06 | 68.81 |
| | Q+AdaRound | 70.16 | 31.97 | 28.24 | 220.01 | 103.15 | 120.37 |
| | Q+FlexRound (Ours) | **66.62** | **31.74** | **27.68** | **145.45** | **101.81** | **106.88** |

Transformer layer that includes attention sublayers and feedforward sublayers. All weights are quantized into 8-bit except the last randomly initialized layer. As for activation quantization, on-the-fly (static) quantization is conducted before every fully-connected layer except the inputs of the softmax layer and the normalization layer that remain to be of full-precision as in Zafrir et al. (2019) and Zhang et al. (2020).

**BERT and GPT-Neo on GLUE** We evaluate the natural language understanding (NLU) performance of FlexRound using various models including BERT$_{Base}$, BERT$_{Large}$, GPT-Neo$_{125M}$, GPT-Neo$_{1.3B}$ and GPT-Neo$_{2.7B}$ on the GLUE benchmark. The learning rate applied to all learnable parameters ($s_1$, $S_2$, and $s_3$) is selected to be 2e-4 for BERT and to be 3e-4 for GPT-Neo. Reconstruction process is performed by using $1024$ random samples for $20K$ iterations. For all experiments, the batch size is 64 and maximum sequence length of all experiments is 128. We utilize pre-trained language models (PLMs) and datasets available from the HuggingFace (Wolf et al., 2020) repository[3]. Further experimental details are referred to Appendix G. In Table 4, we report the performance of 'Q + AdaRound' and 'Q + FlexRound' that are potentially promising as shown in Table 3. We can notice that 'Q + FlexRound' yields better NLU scores than 'Q + AdaRound' for most NLU tasks. In particular, for the MNLI and QQP datasets, 'Q + FlexRound' can achieve comparable or even superior performance to a full-precision model in the per-tensor uniform PTQ setting except GPT-Neo$_{125M}$.

**GPT-Neo and OPT on WikiText2 and PTB** We test the natural language generation (NLG) performance of FlexRound on the WikiText2 and PTB datasets. PLMs (for NLG) are quantized by FlexRound (in a per-tensor quantization manner) while a small amount of data of downstream tasks are used for reconstruction and evaluation. Specifically, PLMs include GPT-Neo$_{125M}$, GPT-Neo$_{1.3B}$, GPT-Neo$_{2.7B}$, OPT$_{125M}$, OPT$_{1.3B}$ and OPT$_{2.7B}$, while 256 downstream task data samples are chosen at random for reconstruction. More details on the experimental setup are provided in Appendix I. Table 5 presents the results of GPT-Neo and OPT on NLG tasks and it is clear that 'Q + FlexRound' is superior to 'Q + AdaRound' for all models and NLG tasks. Note that for GPT-Neo, 'Q + FlexRound' can achieve the similar performance of a full-precision PLM even in the per-tensor uniform PTQ setting, while some previous attempts rely on group-wise or vector-wise quantization (Yao et al., 2022; Dettmers et al., 2022).

## 5 CONCLUSION

We propose a new rounding scheme, named *FlexRound*, for post-training quantization under the the principle of element-wise division, to enable learning both a common quantization grid size and an individual scale for each pre-trained weight. We validate that FlexRound can flexibly quantizes pre-trained weights by exploiting their magnitude as a metric to measure importance. Consequently, FlexRound can achieve comparable performance to a full-precision model even in the per-tensor uniform PTQ setting. As a future work, we plan to quantize large language models beyond 6.7B parameters in the per-tensor uniform PTQ setting.

---

[3]`https://github.com/huggingface/transformers`

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

## A  COMPARISON OF FLEXROUND TO ADAROUND AND ADAQUANT

Figure 4 shows that the comparison of FlexRound to AdaRound and AdaQaunt. As seen in Figure 4(a), FlexRound can quantize pre-trained weights more flexibly than AdaRound and AdaQuant. As weights of large magnitude are not quantized aggressively in the middle of Figure 4(a) compared to the right of Figure 4(a), AdaQuant quantizes weights of large importance marginally, which seems to make it difficult for AdaQuant to quantize MobileNetV2 into 4-bit.

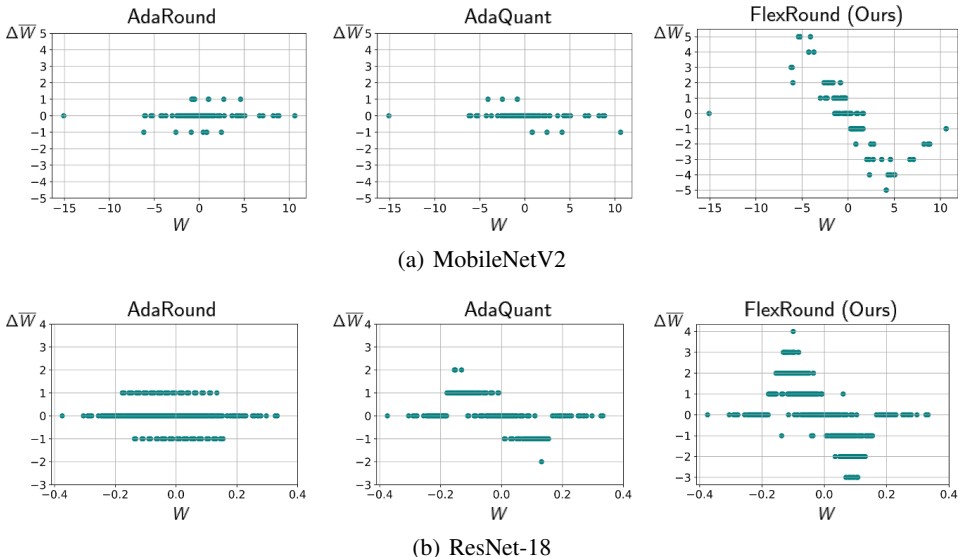

(a) MobileNetV2

(b) ResNet-18

Figure 4: Scatter plot of the amount of grid shifts from rounding-to-nearest gird in the first layer of the first block in MobileNetV2 and ResNet-18 when only weights are quantized into 4-bit.

# B DERIVATION OF SECTION 4.1

Let $\mathcal{L} = \|\boldsymbol{W}\boldsymbol{X} - \widehat{\boldsymbol{W}}\widetilde{\boldsymbol{X}}\|_F^2$ and $\boldsymbol{S}'$ be $\boldsymbol{S}_2 \odot \boldsymbol{s}_3$ for a fully-connected layer and $\boldsymbol{S}_2 \odot \boldsymbol{s}_3 \odot \boldsymbol{s}_4$ for a convolutional layer. In the case of a fully-connected layer,

$$
\begin{aligned}
\frac{\partial \mathcal{L}}{\partial S'_{(i,j)}} &= \frac{\partial \widehat{W}_{(i,j)}}{\partial S'_{(i,j)}} \frac{\partial \mathcal{L}}{\partial \widehat{W}_{(i,j)}} \\
&= \frac{\partial}{\partial S'_{(i,j)}} \left( s_1 \left\lfloor \frac{W_{(i,j)}}{s_1 S'_{(i,j)}} \right\rceil \right) \frac{\partial \mathcal{L}}{\partial \widehat{W}_{(i,j)}} \\
&= s_1 \frac{\partial}{\partial S'_{(i,j)}} \left( \left\lfloor \frac{W_{(i,j)}}{s_1 S'_{(i,j)}} \right\rceil \right) \frac{\partial \mathcal{L}}{\partial \widehat{W}_{(i,j)}} \\
&= s_1 \frac{\partial}{\partial S'_{(i,j)}} \left( \frac{W_{(i,j)}}{s_1 S'_{(i,j)}} \right) \frac{\partial \mathcal{L}}{\partial \widehat{W}_{(i,j)}} \quad (\because \text{Straight-Through Estimator}) \\
&= s_1 \frac{W_{(i,j)}}{s_1} \frac{\partial}{\partial S'_{(i,j)}} \left( \frac{1}{S'_{(i,j)}} \right) \frac{\partial \mathcal{L}}{\partial \widehat{W}_{(i,j)}} \\
&= W_{(i,j)} \left( -\frac{1}{S'^2_{(i,j)}} \right) \frac{\partial \mathcal{L}}{\partial \widehat{W}_{(i,j)}} \\
&= -\frac{W_{(i,j)}}{S'^2_{(i,j)}} \frac{\partial \mathcal{L}}{\partial \widehat{W}_{(i,j)}}
\end{aligned}
$$

The derivation in the case of a convolutional layer can be done by just replacing $\widehat{W}_{(i,j)}$ with $\widehat{W}_{(i,j,k,l)}$ and $S'_{(i,j)}$ with $S'_{(i,j,k,l)}$.

# C   RESNET-18, RESNET-50, AND MOBILENETV2 ON IMAGENET WITH PRE-TRAINED MODELS FROM THE OFFICIAL PYTORCH REPOSITORY

Table 6: Top-1/Top-5 accuracy (%) for ResNet-18, ResNet-50, and MobileNetV2 on ImageNet when only weights are quantized. "B + X" expresses the implementation of X in the BRECQ's setting. We employ pre-trained models available from the official PyTorch repository.

| Method | # Bits (W./A.) | ResNet-18 | ResNet-50 | MobileNetV2 |
|---|---|---|---|---|
| Full-precision | 32/32 | 69.76/89.08 | 76.15/92.87 | 71.88/90.29 |
| B + AdaQuant | 4/32 | 67.55/87.73 | 74.09/91.77 | 0.48/0.53 |
| B + AdaRound | 4/32 | 69.15/88.70 | 75.51/92.73 | 67.76/88.12 |
| B + FlexRound (Ours) | 4/32 | **69.21/88.76** | **75.59/92.63** | **69.56/89.02** |
| B + AdaQuant | 3/32 | 60.75/83.41 | 66.19/87.08 | 0.10/0.52 |
| B + AdaRound | 3/32 | 67.98/88.17 | 74.51/92.20 | 60.18/83.52 |
| B + FlexRound (Ours) | 3/32 | **68.02/88.03** | **74.61/92.11** | **64.85/86.38** |
| B + AdaQuant | 2/32 | 1.13/4.10 | 0.12/0.60 | 0.10/0.50 |
| B + AdaRound | 2/32 | 63.01/85.20 | 68.31/88.98 | 33.10/60.58 |
| B + FlexRound (Ours) | 2/32 | **63.73/85.41** | **70.57/90.07** | **38.09/64.90** |

Table 7: Top-1/Top-5 accuracy (%) for ResNet-18, ResNet-50, and MobileNetV2 on ImageNet when both weights and activations are quantized. "B + X" and "Q + Y" represent the implementation of X in the BRECQ's setting and that of Y in the QDrop's setting, respectively. We employ pre-trained models available from the official PyTorch repository.

| Method | # Bits (W./A.) | ResNet-18 | ResNet-50 | MobileNetV2 |
|---|---|---|---|---|
| Full-precision | 32/32 | 69.76/89.08 | 76.15/92.87 | 71.88/90.29 |
| B + AdaRound | 4/4 | 68.32/88.13 | 74.28/92.02 | 28.46/52.60 |
| B + FlexRound (Ours) | 4/4 | **68.34/88.19** | 74.42/92.04 | 55.25/78.61 |
| Q + AdaRound | 4/4 | 68.19/88.18 | 74.68/92.02 | 56.68/80.95 |
| Q + FlexRound (Ours) | 4/4 | 68.23/88.22 | **74.83/92.11** | **61.56/84.18** |
| B + AdaRound | 3/3 | 64.44/85.73 | 68.80/88.79 | 2.11/7.24 |
| B + FlexRound (Ours) | 3/3 | 64.61/85.85 | 69.62/89.19 | 8.80/21.79 |
| Q + AdaRound | 3/3 | **65.33/86.60** | 71.80/90.72 | 32.41/59.27 |
| Q + FlexRound (Ours) | 3/3 | 65.28/86.49 | **71.84/90.48** | **41.51/68.02** |

# D  IMPORTANCE OF JOINTLY LEARNING THE QUANTIZATION GRID SIZE $s_1$ WITH ROUNDING

Table 8: Top-1/Top-5 accuracy (%) on ImageNet by ResNet-18, ResNet-50, and MobileNetV2 with only weights quantized into 4-bit. "B + X" denotes the implementation of X in the setting of BRECQ. We employ pre-trained models available from the official PyTorch repository.

| Method | ResNet-18 | ResNet-50 | MobileNetV2 |
|---|---|---|---|
| B + AdaQuant | 67.55/87.73 | 74.09/91.77 | 0.48/0.53 |
| B + AdaRound | 69.15/88.70 | 75.51/92.73 | 67.76/88.12 |
| B + FlexRound with $s_1$ fixed | 69.11/88.64 | 75.52/92.64 | 68.19/88.45 |
| B + FlexRound (Ours) | **69.21/88.76** | **75.59/92.63** | **69.56/89.02** |

To demonstrate the importance of jointly learning $s_1$ with the rounding, we did an additional study with $s_1$ fixed. When fixing $s_1$, for ResNet models the performance of FlexRound is almost comparable to that of AdaRound, while for MobileNetV2 FlexRound is somewhat superior to AdaRound. When jointly learning $s_1$ with the rounding, however, FlexRound outperforms AdaRound for all models. It is therefore critical to learn $s_1$ jointly with the rounding.

# E    ABLATION STUDY ON SAMPLE SIZE

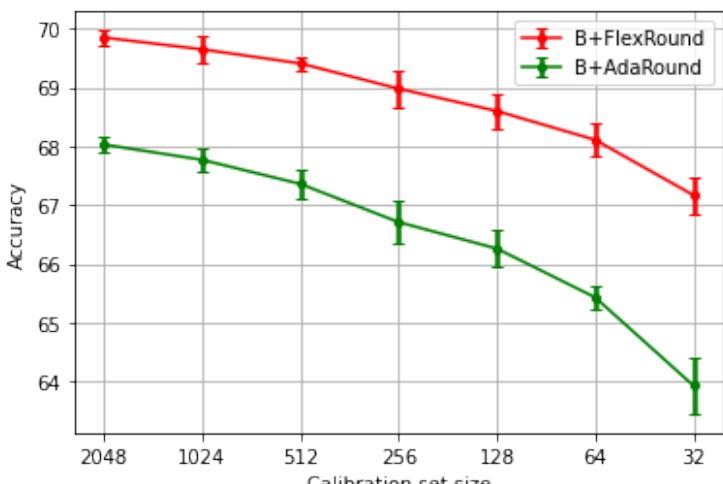

Figure 5: Ablation study on sample size when quantizing MobileNetV2 into 4-bit. Only weights are quantized, with activations kept in full-precision. We employ pre-trained models available from the official PyTorch repository.

No matter how much data is used, B+FlexRound always outperforms B+AdaRound. When the sample size decreases from 64 to 32, the accuracy of B+FlexRound declines by almost one percent. Correspondingly, a sample size of 32 would be a breakthrough point.

# F  COMBINING ELEMENT-WISE ADDITION AND ELEMENT-WISE DIVISION

Table 9: Top-1/Top-5 accuracy (%) for ResNet-18, ResNet-50, and MobileNetV2 on ImageNet when only weights are quantized. "B + X" expresses the implementation of X in the BRECQ's setting. We employ pre-trained models available from the official PyTorch repository.

| Method | # Bits (W./A.) | ResNet-18 | ResNet-50 | MobileNetV2 |
|---|---|---|---|---|
| Full-precision | 32/32 | 69.76/89.08 | 76.15/92.87 | 71.88/90.29 |
| B + AdaQuant | 4/32 | 67.55/87.73 | 74.09/91.77 | 0.48/0.53 |
| B + AdaQuant + FlexRound | 4/32 | 68.75/88.45 | 75.14/92.45 | 68.36/88.49 |
| B + FlexRound (Ours) | 4/32 | **69.21/88.76** | **75.59/92.63** | **69.56/89.02** |
| B + AdaQuant | 3/32 | 60.75/83.41 | 66.19/87.08 | 0.10/0.52 |
| B + AdaQuant + FlexRound | 3/32 | 67.36/87.71 | 74.05/91.87 | 61.64/84.28 |
| B + FlexRound (Ours) | 3/32 | **68.02/88.03** | **74.61/92.11** | **64.85/86.38** |
| B + AdaQuant | 2/32 | 1.13/4.10 | 0.12/0.60 | 0.10/0.50 |
| B + AdaQuant + FlexRound | 2/32 | 62.23/84.77 | 69.39/89.35 | 34.11/61.64 |
| B + FlexRound (Ours) | 2/32 | **63.73/85.41** | **70.57/90.07** | **38.09/64.90** |

To identify whether there comes any benefit from both addition and division, we combine AdaQuant with FlexRound. AdaQuant + FlexRound is superior to AdaQuant but inferior to FlexRound. This might be due to the naive combination of AdaQuant with FlexRound. Considering both addition and division would be an interesting future work.

## G  BERT AND GPT-NEO ON GLUE

The experimental setting of 'Q + AdaRound' follows Wei et al. (2022). To investigate the natural language understanding performance of FlexRound from BERT[4] to GPT-Neo[5], we directly fine-tune pre-trained models on the GLUE[6] dataset. For BERT, we use uncased models. Hyper-parameter selection for fine-tuning a pre-trained model is given in Table 10. We use ADAM optimizer as default for all methods and models. In the QDrop's setting, the probability of dropping activation quantization is set to 0.5. We utilize the Huggingface repository[7] for the evaluation method without any modification.

Table 10: Hyper-parameter selection for fine-tuning $BERT_{Base}$, $BERT_{Large}$, GPT-$Neo_{125M}$, GPT-$Neo_{1.3B}$, and GPT-$Neo_{2.7B}$ on the GLUE benchmark.

| Configuration | $BERT_{Base}$ | $BERT_{Large}$ | GPT-$Neo_{125M}$ | GPT-$Neo_{1.3B}$ | GPT-$Neo_{2.7B}$ |
|---|---|---|---|---|---|
| Learning Rate | 2e-5 | 2e-5 | 2e-5 | 2e-5 | 1e-5 |
| Batch Size | 32 | 32 | 32 | 32 | 16 |
| Epoch | | | 3 | | |
| Max Sequence Length | | | 128 | | |
| Weight Decay | | | 0.01 | | |

---

[4]https://huggingface.co/bert-base-uncased
[5]https://huggingface.co/EleutherAI/gpt-neo-1.3B
[6]https://huggingface.co/datasets/glue
[7]https://github.com/huggingface/transformers/tree/main/examples/pytorch/text-classification

# H  BERT ON SQUAD

Table 11 additionally shows the performace of FlexRound on the SQuADv1(Rajpurkar et al., 2016)[8] dataset for the BERT models. For experimental details, Both BERT$_{Base}$ and BERT$_{Large}$ are uncased models. For 'Q + FlexRound', the learning rate is set to 1e-4 for both models. For both 'Q + AdaRound' and 'Q + FlexRound', the batch size and the number of iterations for reconstruction are $64$ and $20k$ respectively. We use ADAM optimizer as default for all methods and models. The other experimental setting of 'Q + AdaRound' follows Wei et al. (2022). Table 12 shows the hyper-parameter selection for fine-tuning. Both BERT$_{Base}$ and BERT$_{Large}$ are using the same configuration. The other setting for fine-tuning and the evaluation method are the same as HuggingFace repository[9].

Table 11: F1 score for BERT$_{Base}$ and BERT$_{Large}$ on SQuADv1 dataset when both weights and activations are quantized into $8$-bit. "Q + X" represent the implementation of X in the QDrop's setting.

| Method | # Bits (W./A.) | BERT$_{Base}$ | BERT$_{Large}$ |
|---|---|---|---|
| Full-precision | 32/32 | 87.05 | 89.31 |
| Q + AdaRound | 8/8 | 86.90 | 88.89 |
| Q + FlexRound (Ours) | 8/8 | **87.25** | **89.25** |

Table 12: Hyper-parameter selection for fine-tuning BERT$_{Base}$ and BERT$_{Large}$ on SQuADv1 dataset.

| Learning rate | Batch size | Epoch | Maximum sequence length | Document stride |
|---|---|---|---|---|
| 1e-4 | 32 | 4 | 384 | 128 |

---

[8]`https://huggingface.co/datasets/squad`
[9]`https://github.com/huggingface/transformers/tree/main/examples/pytorch/question-answering`

# I   GPT-NEO AND OPT ON WIKITEXT2 AND PTB

To evaluate FlexRound for natural language generation tasks, we utilize the WikiText2 [10] and PTB [11] datasets. Table 13 reports the learning rate, the batch size, and the number of iterations for 'Q + FlexRound'. The experimental setting of 'Q + AdaRound' follows Wei et al. (2022) except the number of iterations; we employ $15k$ iterations for GPT-Neo and $20k$ iterations for OPT[12]. The batch size for 'Q + AdaRound' is same as that for 'Q + FlexRound'. We use ADAM optimizer as default for all methods and models. The probability of dropping activation quantization is set to $0.5$ in the QDrop's setting. We use the Huggingface repository[13] for the evaluation method without any modification.

Table 13: Hyper-parameter selection for 'Q + FlexRound' in Table 5.

| Dataset | Configuration | GPT-Neo$_{125M}$ | GPT-Neo$_{1.3B}$ | GPT-Neo$_{2.7B}$ | OPT$_{125M}$ | OPT$_{1.3B}$ | OPT$_{2.7B}$ |
|---|---|---|---|---|---|---|---|
| WikiText2 | Learning rate | 2e-3 | 6e-4 | 4e-4 | 1e-3 | 9e-5 | 8e-5 |
| | Batch size | 32 | 16 | 8 | 32 | 16 | 8 |
| | Iteration | $15k$ | $15k$ | $15k$ | $5k$ | $5k$ | $5k$ |
| PTB | Learning rate | 4e-3 | 4e-3 | 2e-3 | 8e-4 | 1e-3 | 5e-3 |
| | Batch size | 32 | 16 | 8 | 32 | 16 | 8 |
| | Iteration | $15k$ | $15k$ | $15k$ | $5k$ | $5k$ | $5k$ |

---

[10]https://huggingface.co/datasets/wikitext

[11]https://huggingface.co/datasets/ptb_text_only

[12]https://huggingface.co/facebook/opt-1.3b

[13]https://github.com/huggingface/transformers/tree/main/examples/pytorch/language-modeling

## J    FINETUNED GPT-NEO AND OPT ON WIKITEXT2 AND PTB

As for the evaluation of quantized pre-trained language models, the performance (i.e., accuracy) of quantized OPT (by Q+AdaRound or Q+FlexRound) is not close to that of full-precision OPT, while GPT-Neo can be quantized without noticeable accuracy degradation. To investigate whether such an observation is also valid for finetuned OPT or not, we conduct additional experiments on finetuned OPT and GPT-Neo with Wikitext2 and PTB dataset. As shown in the table 14, quantized model's performance of finetuned OPT turns out to be close to full-precision performance. Considering that the model was finetuned with each downstream dataset, We utilize smaller dataset and lesser iteration for reconstruction. We use 128 samples for calibrations set and the iteration is fixed to 500 for all experiments. Learning rate and batch size for the experiments are shown in Table 15. Other settings are the same as Appendix I.

Table 14: Performance of GPT-Neo$_{125M}$, GPT-Neo$_{1.3B}$, GPT-Neo$_{2.7B}$, OPT$_{125M}$, OPT$_{1.3B}$ and OPT$_{2.7B}$ Finetuned on the WikiText2 and PTB datasets. The perplexity (PPL) is employed as a performance metric. The lower PPL, the better. "Q + X" means the implementation of X in the QDrop's setting.

| Dataset | Method | GPT-Neo$_{125M}$ | GPT-Neo$_{1.3B}$ | GPT-Neo$_{2.7B}$ | OPT$_{125M}$ | OPT$_{1.3B}$ | OPT$_{2.7B}$ |
|---|---|---|---|---|---|---|---|
| WikiText2 | Full-precision | 21.96 | 12.09 | 10.78 | 19.85 | 11.52 | 10.27 |
| | Q+AdaRound | 30.52 | 12.47 | 14.09 | 27.96 | 12.66 | 10.97 |
| | Q+FlexRound (Ours) | **24.30** | **12.37** | **12.43** | **21.43** | **12.02** | **10.63** |
| PTB | Full-precision | 24.20 | 16.09 | 14.70 | 16.50 | 11.62 | 10.80 |
| | Q+AdaRound | 31.40 | 16.63 | 19.80 | 20.28 | 13.00 | 12.02 |
| | Q+FlexRound (Ours) | **26.03** | **16.32** | **16.87** | **17.68** | **12.22** | **11.29** |

Table 15: Hyper-parameter selection for 'Q + FlexRound' in Table 14. Sample size is 128 and iteration is 500.

| Dataset | Configuration | GPT-Neo$_{125M}$ | GPT-Neo$_{1.3B}$ | GPT-Neo$_{2.7B}$ | OPT$_{125M}$ | OPT$_{1.3B}$ | OPT$_{2.7B}$ |
|---|---|---|---|---|---|---|---|
| WikiText2 | Learning rate | 5e-3 | 4e-4 | 4e-3 | 3e-5 | 7e-6 | 1e-5 |
| | Batch size | 32 | 16 | 8 | 32 | 16 | 8 |
| PTB | Learning rate | 5e-3 | 7e-3 | 7e-3 | 5e-5 | 3e-5 | 8e-6 |
| | Batch size | 32 | 16 | 8 | 32 | 16 | 8 |

