# OpenReview forum: "FlexRound: Learnable Rounding by Element-wise Division for Post-Training Quantization"
_ICLR.cc/2023/Conference — Submitted to ICLR 2023_

### Official Review · Reviewer_y3vc · 2022-10-23

**Confidence:** 3
**Correctness:** 3
**Technical Novelty And Significance:** 3
**Empirical Novelty And Significance:** 3
**Recommendation:** 5

**Clarity, Quality, Novelty And Reproducibility:**

* Clarity:

  * I found section 3.3 (Leveraging the Importance of Pre-Trained Weights) confusing. As it is part of section 3
  (Methodology), one would expect that it describes some part of the algorithm. Furthermore, by the use of the word
  "Leveraging", one would expect that the weight magnitude is somehow explicitly used to alter the rounding or scaling
  policy introduced in the previous section (equation 2). It turns out (if my understanding is correct), that section
  3.3 only shows an analysis on the rounding scheme resulting from experiments on MobileNetV2 and ResNet-18 using the
  algorithm presented in section 3.2. I think this analysis should be moved to the experimental results section.
  * It is not clear whether baselines have been modified to also work in a per-tensor PTQ reconstruction setup (this
  may explain the difference with reported results in the original paper for BRECQ).

* Novelty:

  * Up to my knowledge, the formulation is technically novel and sufficiently different from previously introduced
  techniques.

* Quality:

  * Similar to the range equalization method presented in (Nagel et al., 2019), this method attempts
  to re-scale weights, albeit during quantization and not prior to it. While not applicable to all architectures,
  weight pre-scaling is fast and works with any quantization algorithm. However, no experiment is run to see how this
  method compares to this baseline, and whether using pre-scaled weights as input to this method further improves
  results.

* Reproducibility:

  * The authors provide sufficient detail to reproduce the experiments and express their intention to make code publicly
  available.

**Strength And Weaknesses:**

* Strengths:

  * The main idea is simple and attractive from both a theoretical and practical standpoint, extending neatly the
  existing algorithmic frameworks (e.g. BRECQ and QDrop)
  * Evaluation scope covers a wide variety of domains (language, vision) and architectures (ResNet, MobileNet, ViT).

* Weaknesses:

  * There are points where the paper is unclear. Please see comments on clarity below.
  * The experimental results reported for BRECQ in table 3 are different from those reported in the original papers.
  I presume this is because a per-tensor quantization comparison is being used (but the authors should confirm this).
  * No study in conjunction with the related weight range equalization method introduced in
  (Nagel et al., 2019) is shown, nor is a comparison carried out. Please see comments below on quality.

**Summary Of The Paper:**

The paper proposes a post-training quantization (PTQ) weight/activation quantization method.
The gist is to simultaneously optimize a multiplicative element-wise scale factor applied to the target tensor
and the quantization step size, under the reconstruction loss. The method can be seen as closely related in spirit
to AdaQuant and AdaRound, with the critical difference that instead of adding optimal noise to the tensor before
rounding, multiplicative noise is used instead. If can also be seen to be related to the weight scaling method of
(Nagel et al, 2019), in the sense that weight tensors are scaled prior to rounding, but with the critical difference
that DFQ does this prior to quantization, while FlexQuant does it during the quantization step.

Experiments carried out on image classification (ImageNet), natural language understanding (GLUE) and natural language
generation (WikiText2 and PTB) show superior performance to AdaRound and AdaQuant.

**Summary Of The Review:**

I think this work provides a simple yet effective weight scaling approach within the PTQ framework, and,
up to my knowledge, the problem formulation is qualitatively different from previous approaches. While evaluation
is broad in terms of the benchmarks used, I have two concerns. First, the results reported for the
BRECQ + AdaRound baselines are different from the ones in the original paper. Second, I would have liked to also see a
comparison/combination with the weight equalization approach of (Nagel et al., 2019). The clarity of the presentation
could be improved.

---

> ### Author Response · Authors · 2022-11-12
> **Dear Reviewer y3vc**
>
> Dear Reviewer y3vc,
>
> We appreciate your insightful comments.
>
> ----------------------------------------
>
> $\textbf{[Q1. Mismatch between our results and the results in the original papers]}$
>
> As explained in the general response above, there are two main differences between our results and the results in the original paper. First of all, we used pre-trained models available in the official PyTorch repository whereas the authors of BRECQ and QDrop have designed their own pre-trained models (that significantly outperform those from the official PyTorch repository). Besides, as the reviewer guessed, we adopt $\textit{per-tensor}$ linear $\textit{symmetric}$ weight quantization while the authors of BRECQ and QDrop adopt $\textit{per-channel}$ linear $\textit{asymmetric}$ weight quantization. When using pre-trained models available from the BRECQ github repository, we could obtain the results more similar to those in the original BRECQ and QDrop papers as we reproduced (shown in the general response).
>
> ----------------------------------------
>
> $\textbf{[Q2. Section 3.3 should be moved to Section 4]}$
>
> Thank you for the helpful suggestion.
> Following your suggestion, we will move Section 3.3 to Section 4 in the revision.
>
> ----------------------------------------
>
> $\textbf{[Q3. Using pre-scaled weights as a baseline model]}$
>
> We appreciate your constructive suggestion.
> It is known that preprocessing weights through the range equalization (cross layer equalization (CLE) and/or absorbing high biases (AHB)) method exhibits a noticeable enhancement for the per-tensor quantization performance in vision models (Nagel et al., 2019, Nagel et al., 2021). We agree with the reviewer's insight and hence conducted an additional experiment as follows. In all experiments, we used the same configuration as the previous setting (Table 2).
>
> Experiment of the 4-bit MobileNetV2 pre-processed using cross layer equalization (CLE) and absorbing high biases (AHB)
> | Method | Top-1 Accuracy |
> |:----------|:------------:|
> | B+AdaRound | 67.76 |
> | B+AdaRound + CLE + AHB | $\mathbf{70.04}$ |
> | B+FlexRound | 69.56 |
> | B+FlexRound + CLE + AHB | 69.74 |
>
> In the MobileNetV2 experiment using a model which is pre-processed with cross-layer equalization and absorbing high biases, B+AdaRound shows better accuracy (compared to the case of non-pre-processed model, and can outperform FlexRound.
> In a language model based on the transformer architecture, however, it is not known how to apply CLE to a transformer-based architecture. Consequently, we naively apply CLE only to the feed-forward module of BERT Base. All experiments follow the same configuration of the previous setting (Table 4).
>
> Experiment of the 8/8-bit BERT-base-uncased preprocessed using CLE (only feed-forward modules)
> | Method | MNLI Accuracy (matched / mis-matched) |
> |:----------|:------------:|
> | FP | 84.49 / 85.20 |
> | FP + CLE | 84.49 / 85.19 |
> | Q + AdaRound |83.69 / 84.61 |
> | Q + AdaRound + CLE | 83.94 / 84.60 |
> | Q + FlexRound | $\mathbf{84.53 / 84.98}$ |
> | Q + FlexRound + CLE  | 84.28 / 85.00 |
>
> As shown in the table above, CLE slightly improves Q+AdaRound while Q+FlexRound does not match with CLE preprocessing. Nonetheless, Q+FlexRound surpasses the others. As the reviewer mentioned, the range equalization is not applicable to all architectures, and it is uncertain about how to apply CLE to transformer-based architectures. On the other hand, FlexRound can quantize weights well even in the form of an end-to-end configuration, regardless of what kind of architecture FlexRound chooses to quantize. We thus believe that FlexRound would have its own advantages compared to other post-training weight quantization methods (that might need preprocessing for better performance).
>
> Nagel et al., 2019. Data-Free Quantization Through Weight Equalization and Bias Correction.
> Nagel et al., 2021. A White Paper on Neural Network Quantization.
>
> --------------------------------------------------------
>
> Thank you for your insightful comment again. We will also provide the language model code soon.

---

### Official Review · Reviewer_vGda · 2022-10-24

**Confidence:** 3
**Clarity, Quality, Novelty And Reproducibility:** 1. There are some confusing points, p…
**Correctness:** 3
**Technical Novelty And Significance:** 2
**Empirical Novelty And Significance:** 2
**Recommendation:** 5

**Details Of Ethics Concerns:**

N.A.

**Strength And Weaknesses:**

Weaknesses:
1. As the author mention that the previous works used element-wsie addition in parameters refine before quantization while the proposed method used division, why addition is not considered in the submission? Since both division and addition can refine parameters, would it be better if both are used?
2. It seems author miss mentioning how much data is necessary to learn $\boldsymbol{S}$.

**Summary Of The Paper:**

This submission proposed to conduct post-training quantization by learning element-wise scaling parameters $\boldsymbol{S}$. Pre-quantized parameters are divided by the scaling parameters and then rounded to integer. Basically, the proposed method is a subtle re-training for pre-trained parameters for quantization.

**Summary Of The Review:**

Basically, the core method can be considered as a subtle re-training for pre-trained parameters for quantization. Besides, the author missed some important points for precise evaluation.

---

> ### Author Response · Authors · 2022-11-12
> **Dear Reviewer vGda**
>
> Dear Reviewer vGda,
>
> We appreciate your constructive comments.
>
> ------------
>
> $\textbf{[Q1. Combine an additive approach with a division-based approach]}$
>
> Thank you for the interesting suggestion. Although we agree that it would be interesting to combine AdaRound and FlexRound, we are afraid that it would be challenging due to the fact that AdaRound cannot learn $s_1$ jointly with the rounding. Alternatively, we combine AdaQuant with FlexRound since AdaQuant can learn $s_1$ jointly with the rounding. The following table shows the result of combining AdaQuant with FlexRound.
>
> | Method | \# Bits (W./A.) | ResNet-18 | ResNet-50 | MobileNetV2 |
> |:----------|:-----------------:|:--------------:|:--------------:|:----------------:|
> | Full-precision | 32 / 32 | 69.76 / 89.08 | 76.15 / 92.87 | 71.88 / 90.29 |
> | B + AdaQuant | 4 / 32 | 67.55 / 87.73 | 74.09 / 91.77 | 0.48 / 0.53 |
> | B + AdaQuant + FlexRound | 4 / 32 | 68.75 / 88.45 | 75.14 / 92.45 | 68.36 / 88.49 |
> | B + FlexRound (Ours) | 4 / 32 | $\mathbf{69.21} / \mathbf{88.76}$ | $\mathbf{75.59} / \mathbf{92.63}$ | $\mathbf{69.56} / \mathbf{89.02}$ |
> | B + AdaQuant | 3 / 32 | 60.75 / 83.41 | 66.19 / 87.08 | 0.10 / 0.52 |
> | B + AdaQuant + FlexRound | 3 / 32 | 67.36 / 87.71 | 74.05 / 91.87 | 61.64 / 84.28 |
> | B + FlexRound (Ours) | 3 / 32 | $\mathbf{68.02} / \mathbf{88.03}$ | $\mathbf{74.61} / \mathbf{92.11}$ | $\mathbf{64.85} / \mathbf{86.38}$ |
> | B + AdaQuant | 2 / 32 | 1.13 / 4.10 | 0.12 / 0.60 | 0.10 / 0.50 |
> | B + AdaQuant + FlexRound | 2 / 32 | 62.23 / 84.77 | 69.39 / 89.35 | 34.11 / 61.64 |
> | B + FlexRound (Ours) | 2 / 32 | $\mathbf{63.73} / \mathbf{85.41}$ | $\mathbf{70.57} / \mathbf{90.07}$ | $\mathbf{38.09} / \mathbf{64.90}$ |
>
> AdaQuant + FlexRound is superior to AdaQuant but inferior to FlexRound. This might be due to the naive combination of AdaQuant with FlexRound. Considering both addition and division would be an interesting future work.
>
> ------------------------------------
>
> $\textbf{[Q2. Ablation study on how much data is needed to learn $\mathbf{S}$]}$
>
> Thank you for the constructive comment. The following table shows the result of B+AdaRound and B+FlexRound when the sample size varies.
>
> | Sample size | 2048 | 1024 | 512 | 256 | 128 | 64 | 32 |
> |:---------------|:-------:|:-------:|:------:|:------:|:-----:|:-----:|:----:|
> | B + AdaRound | $68.03 \pm 0.13$ |  $67.77 \pm 0.21$ | $67.36 \pm 0.25$ | $66.71 \pm 0.35$ | $66.26 \pm 0.32$ | $65.43 \pm 0.21$ | $63.93 \pm 0.48$ |
> | B + FlexRound | $\mathbf{69.85 \pm 0.12}$ | $\mathbf{69.65 \pm 0.24}$ | $\mathbf{69.41 \pm 0.12}$ | $\mathbf{68.98 \pm 0.32}$ | $\mathbf{68.60 \pm 0.30}$ | $\mathbf{68.11 \pm 0.27}$ | $\mathbf{67.17 \pm 0.31}$ |
>
> No matter how much data is used, B+FlexRound always outperforms B+AdaRound. When the sample size decreases from 64 to 32, the accuracy of B+FlexRound declines by almost one percent. Correspondingly, a sample size of 32 would be a breakthrough point. We will add this result in the appendix of the revision.
>
> ------------------------------------

---

> > ### Comment · Reviewer_vGda · 2022-11-29
> > **Response to Authors**
> >
> > Thanks for authors' response and addition experiments.
> >
> > According to the extra experiment results, it seems that simply combining addition and division in scaling factor learning does not provide improvement, or even worsen. It may contribute to that learning addition factor using scarse data is difficult, compared to division counterpart. As a result, simply changing addition to division (as conducted and proposed in this submission) improves PTQ.
> >
> > Experiments in part 2 demonstrates that roughly 512 instances is able to reach a reasonable performance, which is promissing.
> >
> > Overall, the submission empiricially find that learning division factor on a pre-trained model using scrase data is sufficient to produce a quantized counterpart.
> >
> > I keep my overall score.

---

> > > ### Author Response · Authors · 2022-12-01
> > > **Dear Reviewer vGda**
> > >
> > > Dear Reviewer vGda,
> > >
> > > We sincerely appreciate your time and efforts in your detailed response.
> > >
> > > If there are ways to increase your score, please let us know.
> > >
> > > We would be happy to address any other remaining concerns or discuss any topics if you suggest.

---

### Official Review · Reviewer_mQzD · 2022-10-24

**Confidence:** 4
**Clarity, Quality, Novelty And Reproducibility:** See the above.
**Correctness:** 3
**Technical Novelty And Significance:** 3
**Empirical Novelty And Significance:** 3
**Recommendation:** 5

**Strength And Weaknesses:**

Strengths:

- The authors propose FlexRound, a new approach that applies element-wise division to exploit the parameter magnitude to update the quantization scale. Figure 3 and Figure 4 provide good explanations on the advantages of FlexRound.

- The authors conduct comprehensive empirical studies over the image classification, natural language understanding and natural language generation tasks. Moreover, the authors testify various network architectures across CV and NLP, and especially, GPT-Neo and OPT, for post-training quantization.

- The writing is clean.

Weakness:

- Lack of comparisons with existing approaches, especially BRECQ and Q-Drop. According to the published results of BRECQ and Q-Drop, the results in this paper seem to be worse than these two baselines (e.g., W3-A3 Q-Drop for MobileNet-v2 is 57.98%, but only 41.51% in this paper), even under the same configuration. More comparisons and explanations should be presented.

- While BRECQ and AdaRound rely on element-wise addition for quantization, it is still unclear what is their disadvantage according to the paper.

- The proposed approach seems to be only applicable to weight quantization, yet this is hardly mentioned. For Table-3, it should be made clear that what type of activation quantization is adopted.



Detailed comments:

- Despite ablated, it is still not clear the benefits of introducing $\mathbf{s}_3$ and $\mathbf{s}_4$ in Equation 2. Intuitively, both factors can be absorbed into $\mathbf{S}_2$.

- What is the initialization method for $\mathbf{S}$? This should be important as PTQ does not allow intensive training like those in LSQ.

- Provide more derivations (in appendix) for the gradient w.r.t. the scaling factor $\mathbf{S}^{'}$.

- As a minor suggestion, a loss curve or accuracy curve to compare FlexRound with BRECQ during the PTQ iterations could help illustrate FlexRound. This is based on the intuition that parameters with larger magnitude tend to move faster and thus converge better given the limited training iterations and samples.

- Some recent research on PTQ for your reference.

  - Liu Z, Wang Y, Han K, et al. Post-training quantization for vision transformer. NeurIPS 2021.

  - Bai H, Hou L, Shang L, et al. Towards efficient post-training quantization of pre-trained language models. NeurIPS 2022.

  - Park G, Park B, Kwon S J, et al. nuQmm: Quantized MatMul for Efficient Inference of Large-Scale Generative Language Models. arXiv preprint arXiv:2206.09557, 2022.






**Summary Of The Paper:**

This paper proposes a new approach named FlexRound to improve the post-training quantization of deep neural networks. Previous methods primarily rely on element-wise addition for the rounding function. The proposed FlexRound, instead, uses element-wise division, which can exploit the parameter magnitude to update the quantization scale. The authors validate the efficacy of FlexRound over various CV and NLP tasks, together with a variety of network architectures.

**Summary Of The Review:**

Despite the methodological change is minor, the authors provide a reasonable explanation why FlexRound can exploit the importance of pre-trained weights to better calibrate the quantized model (Figure 3 and Figure 4). Experimentally, I appreciate the authors provide comprehensive studies across computer vision, natural language understanding and natural language generation tasks, and results demonstrate the success. Nonetheless, there are still issues with the experimental designs and comparisons, as mentioned above. There are also concerns w.r.t. the methodology, e.g., if FlexRound can be applied to activation quantization; and the necessity in introducing multiple scaling factors $\mathbf{s}3$, $\mathbf{s}4$, e.t.c..

---

> ### Author Response · Authors · 2022-11-12
> **Dear Reviewer mQzD**
>
> Dear Reviewer mQzD,
>
> We appreciate your constructive feedback.
>
> $\textbf{[Q1. Mismatch between our results and the results in the original papers]}$
>
> As explained in the general response above, there are two main differences between our results and the results in the original paper. First of all, we used pre-trained models available in the official PyTorch repository whereas the authors of BRECQ and QDrop have designed their own pre-trained models (that significantly outperform those from the official PyTorch repository). Besides, we adopt $\textit{per-tensor}$ linear $\textit{symmetric}$ weight quantization while the authors of BRECQ and QDrop adopt $\textit{per-channel}$ linear $\textit{asymmetric}$ weight quantization. When using pre-trained models available from the BRECQ github repository, we could obtain the results more similar to those in the original BRECQ and QDrop papers as we reproduced (shown in the general response).
>
> ------------------------------------------------------------------
>
> $\textbf{[Q2. Disadvantage of AdaRound/BRECQ]}$
>
> As the reviewer oCmz (the first reviewer) mentioned, AdaRound/BRECQ cannot learn $s_1$ jointly with the rounding, which seems to be an obvious disadvantage. In addition, AdaRound/BRECQ allows for rounding either up or down only at most, whereas FlexRound can quantize weights adaptively.
>
> ------------------------------------------------------------------
>
> $\textbf{[Q3. FlexRound seems to be only applicable to weight quantization. What kind of activation quantization is used?]}$
>
> As the reviewer pointed out, FlexRound is designed for weight quantization, which is also the case with AdaRound and AdaQuant. To avoid any unnecessary ambiguity, we will clarify that FlexRound is a post-training weight quantization method in the revised version.
> If we apply FlexRound to activation quantization, a huge overhead in memory would occur since activations need to be dynamically divided by $\mathbf{S}_2$. Thus, similar to AdaRound and AdaQuant again, FlexRound would not be appropriate for activation quantization either.
> As described in the first paragraph of Section 4, we adopt LSQ for activation quantization which is also introduced in BRECQ and QDrop.
>
> ------------------------------------------------------------------
>
> $\textbf{[Q4. Benefit of introducing s3 and s4]}$
>
> Even though  $\mathbf{s}_3$ and $\mathbf{s}_4$ can be absorbed into $\mathbf{S}_2$, as elaborated in the third paragraph of Section 3.2, we additionally introduce $\mathbf{s}_3$ and $\mathbf{s}_4$ to take into account the variation of output channel’s statistics and that of input channel’s statistics. Such a consideration can be empirically supported such that introducing $\mathbf{s}_3$ and $\mathbf{s}_4$ separately is beneficial as shown in Table 1.
>
> ------------------------------------------------------------------
>
> $\textbf{[Q5. Initialization method for S]}$
>
> We initialize all entries of $\mathbf{S}_2$, $\mathbf{s}_3$, and $\mathbf{s}_4$ to be ones in order to enable learning $\mathbf{S}_2$, $\mathbf{s}_3$, and $\mathbf{s}_4$ from rounding-to-nearest, $s_1 \Big\lfloor {\mathbf{W} \over s_1} \Big\rceil$. We will add it in the revision.
>
> ------------------------------------------------------------------
>
> $\textbf{[Q6. Derivation of ${{\partial\mathcal{L}} \over {\partial \mathbf{S'}}}$]}$
>
> Thank you for the helpful comment. We will add it in the appendix of the revised version.
>
> ------------------------------------------------------------------
>
> $\textbf{[Q7. Comparison of loss or accuracy curve between BRECQ and FlexRound]}$
>
> Thank you for the interesting suggestion. Even though we agree that it would be helpful to compare a loss or accuracy curve between BRECQ and FlexRound, unfortunately, there are some issues in our opinion. First, FlexRound considers only the reconstruction error whereas BRECQ considers not only the reconstruction error but also the regularization term to decide rounding up or down. Additionally, when implementing BRECQ, the regularization term is added during minimizing the reconstruction error, not from the beginning. Thus, we feel that It would be difficult to fairly compare a loss or accuracy curve between BRECQ and FlexRound. If you have any interesting ideas for experiments to demonstrate your intuition, please feel free to suggest.
>
> ------------------------------------------------------------------
>
> $\textbf{[Q8. Some recent papers on PTQ]}$
>
> Thank you for the helpful suggestion. We will add them in the revision.
>
> ------------------------------------------------------------------

---

### Official Review · Reviewer_oCmz · 2022-10-24

**Confidence:** 4
**Correctness:** 3
**Technical Novelty And Significance:** 2
**Empirical Novelty And Significance:** 3
**Recommendation:** 6

**Clarity, Quality, Novelty And Reproducibility:**

The paper is clearly written and easy to follow. It has some ablation studies on empirical choices (like s3/s4) which I appreciated. The proposed method is fairly simple and a closely related to AdaQuant and AdaRound, though somewhat novel as the additional factor is a division instead of an addition as in prior work. The paper seems that it should be reproducible but there are some questions wrt the comparisons to prior work (AdaRound and BRECQ, see above).

**Strength And Weaknesses:**


Strength:
* The paper is well written and easy to follow. The visualizations are nice and helpful and the work is put well in context of existing literature.
* FlexRound allows learning the scale and ‘rounding’ at the same time which can potentially be an advantage over SOTA current methods (AdaRound/BRECQ).
* The proposed method is extensively evaluated on image classification and NLP tasks and shows good empirical performance (except some missing comparisons, more on that later).
* Has ablation study on empirical choices such as introducing the factors s3 and s4 which should be in theory not needed.

Weaknesses:
* A proper comparison to literature and prior work is missing. While table 2 should in theory be comparable to the papers, their numbers do not match. To my understanding,’ B+AdaRound’ should be exactly what BRECQ is, however, the stated numbers are significantly below the results in the original BRECQ paper. Where does the difference come from? Also the 4-bit MobileNet v2 results are below the AdaRound paper while actually BRECQ+AdaRound improves over vanilla AdaRound (and the original AdaRound paper uses per-tensor quantization and the first/last layer are in 4 bits).
* Claim 3 states that they are the first that do extensive per-tensor study on image classification and NLP. This is not fully true. The white paper of Nagel et al. 2021 (which the authors also cite) has in its PTQ chapter (table 6) a similarly extensive study which also includes per-tensor quantization. On the per-tensor vs per-channel point later, the original AdaRound paper is also with per-tensor quantization (not sure about BRECQ).
* Claim 2 says they demonstrate that element-wise division includes the importance of the pre-trained weight. This is a fairly strong claim and IMO they only show this partly. They show that the gradient is proportional to the magnitude of the weights but then the link to importance is a bit soft/vague.
* The authors argue that addition schemes may change the sign of a weight. However, for the most compared schemes, AdaRound and BRECQ, this can not happened based on how it is defined. Only in the case of AdaQuant, which they show performs poorly, this could theoretically happen.
* Arguing that FlexRound has no extra hyper-parameters (compared to BRECQ/AdaRound) is only somewhat a benefit as the AdaRound paper keeps al hyper-parameter constant inter experimentation and is therefore de-facto also hyper-parameter free.

Question:
* It seems s1 (the general scaling factor) is learned jointly with the ‘rounding’. Could this be also a reason why FelxRound is empirically better than AdaRound/BRECQ? Due to their formulation, AdaRound/BRECQ can not learn the scales jointly with the rounding which is a clear drawback. An additional ablation (in table 1) with a fixed s1 could potentially give some interesting insights into this.
* Did the author explore combining the additive approach (AdaRound/BRECQ) with the devision based approach? Given that empirically they need 3 new learnable scaling factors (s2, s3, s4,  cf table 1), it might be interesting to see if there comes benefits from both and additive and multiplicative term.


Editorial:
* Would suggest to use $\lfloor \cdot \rceil$ for rounding such as in most prior literature (BRECQ, AdaRound etc).
* Section 3.2: “per-tensor quantization schemes facilitate higher parallelism for implementation compared to per-channel quantization schemes (Nagel et al., 2021)”. This seems a not widely known/acknowledged statement and I could not find such a statement in the referred work. Could the authors please point me to the sections where this is discussed or other resources?



**Summary Of The Paper:**

The paper introduces a new post-training quantization algorithm called FlexRound. Unlike prior algorithms such as AdaRound, the learnable rounding factors are division parameters instead of an addition parameters. It has the advantage that it can learn the integer scale and rounding jointly, and also take the weights ‘importance’ into account according to the authors. The proposed approach is well evaluated on CV and NLP tasks and shows good empirical performance compared to their baselines.


**Summary Of The Review:**

The paper proposes a simple adaptation to AdaRound/BRECQ/AdaQuant which has some novelty as the learned parameter is relative instead of additive. In general the paper has a good empirical evaluation except a few questions/inconsistencies with respect to prior work. A few of the papers claims are a bit strong or can be misleading and should be adapted accordingly. Overall the paper is borderline.

---

> ### Author Response · Authors · 2022-11-12
> **Dear Reviewer oCmz [1]**
>
> Dear Reviewer oCmz,
>
> We appreciate your constructive feedback.
>
> $\textbf{[Q1. Mismatch between our results and the results in the original papers]}$
>
> As discussed in the general response above, for BRECQ results, there are two main differences between our manuscript and the original paper. First, we used pre-trained models available in the official PyTorch repository whereas the authors of BRECQ and QDrop have designed their own pre-trained models (that significantly outperform the models from the official PyTorch repository). Besides, we adopt $\textit{per-tensor}$ linear $\textit{symmetric}$ weight quantization while the authors of BRECQ and QDrop adopt $\textit{per-channel}$ linear $\textit{asymmetric}$ weight quantization. When using pre-trained models available from the BRECQ github repository, we could obtain the results more similar to those in the original BRECQ and QDrop papers as we reproduced (shown in the general response).
> As the reviewer pointed out, in the case of the 4-bit MobileNetV2, the result of vanilla AdaRound in the AdaRound paper is better than our result (indicated as ‘B + AdaRound’) because 1) the authors of the AdaRound paper used 2048 samples while we used 1024 samples and 2) more importantly, the authors of the AdaRound paper choose cross-layer range equalization (Nagel et al., 2019) as a preprocessing while we do not. If we conduct an experiment of ‘B + AdaRound’ in the case of the 4-bit MobileNet by employing cross-layer range equalization as a preprocessing in our experimental setting, we also obtain 70.04 top-1 accuracy which is higher than the result of vanilla AdaRound in the AdaRound paper. As such, we firmly believe that the main difference between our result of ‘B + AdaRound’ and the result of vanilla AdaRound in the AdaRound paper is induced by a selection of cross-layer range equalization as preprocessing.
>
> Nagel et al., 2019. Data-Free Quantization Through Weight Equalization and Bias Correction.
>
> ------------------------------------------------------------------------
>
> $\textbf{[Q2. Not the first per-tensor study on image classification and NLP]}$
>
> As the reviewer mentioned, the authors of the original AdaRound paper did per-tensor study on image classification, and the authors of the white paper did per-tensor study on natural language understanding. However, all of them have not investigated per-tensor study on $\textit{natural language generation}$. Correspondingly, we will revise our manuscript to express “we are the first to conduct extensive experiments in the form of per-tensor uniform PTQ reconstruction on natural language generation as well as image classification and natural language understanding”.
>
> ------------------------------------------------------------------------
>
> $\textbf{[Q3. Link between magnitude and importance]}$
>
> As the reviewer pointed out, the link between magnitude and importance can be soft and/or vague. Nonetheless, we feel that the importance of weights is originally a vaguely defined concept as mentioned in Lee et al., 2021. Accordingly, we believe that there are various views on how to measure the importance of weights. Inspired by the works of Han et al., 2015 and Zhu & Gupta, 2017 that prune unimportant weights based on their magnitudes, the magnitude of weights can be viewed as an importance score.
>
> Lee et al., 2021. Layer-adaptive Sparsity for the Magnitude-based Pruning.
> Han et al., 2015. Learning both weights and connections for efficient neural network.
> Zhu & Gupta, 2017. To prune, or not to prune: exploring the efficacy of pruning for model compression.
>
> ------------------------------------------------------------------------
>
> $\textbf{[Q4. The sign change of a weight can happen only in the case of AdaQuant]}$
>
> As the reviewer pointed out, only AdaQuant may change the sign of a weight, which can be readily observed in the original AdaQuant paper. In the revised manuscript, we clarify that only AdaQuant may change the sign of a weight.
>
> ------------------------------------------------------------------------
>
> $\textbf{[Q5. AdaRound is de-facto hyper-parameter free]}$
>
> We agree with the reviewer’s comment. As the AdaRound paper keeps almost all hyper-parameters fixed, we will remove the expression that AdaRound requires many hyper-parameters in the revised version.
>
> ------------------------------------------------------------------------

---

> > ### Author Response · Authors · 2022-11-12
> > **Dear Reviewer oCmz [2]**
> >
> > $\textbf{[Q6. Ablation study on FlexRound with fixed s1]}$
> >
> > Thank you for the helpful suggestion. We conducted an ablation study on FlexRound with $s_1$ fixed. The following table shows the results when only weights are quantized into 4-bit.
> >
> > | Method | ResNet-18 | ResNet-50 | MobileNetV2 |
> > |:----------|:--------------:|:--------------:|:----------------:|
> > | B + AdaQuant | 67.55 / 87.73 | 74.09 / 91.77 | 0.48 / 0.53 |
> > | B + AdaRound | 69.15 / 88.70 | 75.51 / 92.73 | 67.76 / 88.12 |
> > | B + FlexRound with $s_1$ fixed | 69.11 / 88.64 | 75.52 / 92.64 | 68.19 / 88.45 |
> > | B + FlexRound (Ours) | $\mathbf{69.21} / \mathbf{88.76}$ | $\mathbf{75.59} / \mathbf{92.63}$ | $\mathbf{69.56} / \mathbf{89.02}$ |
> >
> > When fixing $s_1$, for ResNet models the performance of FlexRound is almost comparable to that of AdaRound, while for MobileNetV2 FlexRound is somewhat superior to AdaRound. When jointly learning $s_1$ with the rounding, however, FlexRound outperforms AdaRound for all models. It is therefore critical to learn $s_1$ jointly with the rounding.
> >
> > ------------------------------------------------------------------------
> >
> > $\textbf{[Q7. Combination of an additive approach with a division-based approach]}$
> >
> > Thank you for the interesting suggestion. We agree that it would be interesting to combine AdaRound and FlexRound. However, please understand that if we combine AdaRound with FlexRound, such a combination might be challenging due to the fact that AdaRound cannot learn $s_1$ jointly with the rounding. Alternatively, we combine AdaQuant with FlexRound since AdaQuant can learn $s_1$ jointly with the rounding. The following table shows the result of combining AdaQuant with FlexRound.
> >
> > | Method | \# Bits (W./A.) | ResNet-18 | ResNet-50 | MobileNetV2 |
> > |:----------|:-----------------:|:--------------:|:--------------:|:----------------:|
> > | Full-precision | 32 / 32 | 69.76 / 89.08 | 76.15 / 92.87 | 71.88 / 90.29 |
> > | B + AdaQuant | 4 / 32 | 67.55 / 87.73 | 74.09 / 91.77 | 0.48 / 0.53 |
> > | B + AdaQuant + FlexRound | 4 / 32 | 68.75 / 88.45 | 75.14 / 92.45 | 68.36 / 88.49 |
> > | B + FlexRound (Ours) | 4 / 32 | $\mathbf{69.21} / \mathbf{88.76}$ | $\mathbf{75.59} / \mathbf{92.63}$ | $\mathbf{69.56} / \mathbf{89.02}$ |
> > | B + AdaQuant | 3 / 32 | 60.75 / 83.41 | 66.19 / 87.08 | 0.10 / 0.52 |
> > | B + AdaQuant + FlexRound | 3 / 32 | 67.36 / 87.71 | 74.05 / 91.87 | 61.64 / 84.28 |
> > | B + FlexRound (Ours) | 3 / 32 | $\mathbf{68.02} / \mathbf{88.03}$ | $\mathbf{74.61} / \mathbf{92.11}$ | $\mathbf{64.85} / \mathbf{86.38}$ |
> > | B + AdaQuant | 2 / 32 | 1.13 / 4.10 | 0.12 / 0.60 | 0.10 / 0.50 |
> > | B + AdaQuant + FlexRound | 2 / 32 | 62.23 / 84.77 | 69.39 / 89.35 | 34.11 / 61.64 |
> > | B + FlexRound (Ours) | 2 / 32 | $\mathbf{63.73} / \mathbf{85.41}$ | $\mathbf{70.57} / \mathbf{90.07}$ | $\mathbf{38.09} / \mathbf{64.90}$ |
> >
> > AdaQuant + FlexRound is superior to AdaQuant but inferior to FlexRound. This might be due to the naive combination of AdaQuant with FlexRound. Considering both addition and division would be an interesting future work.
> >
> > ------------------------------------------------------------------------
> >
> > $\textbf{[Q8. Suggestion about using ⌊⋅⌉ for rounding]}$
> >
> > We appreciate your suggestion. We replace [⋅] with ⌊⋅⌉ in the revision.
> >
> > ------------------------------------------------------------------------
> >
> > $\textbf{[Q9. Higher parallelism implementation than per-channel quantization]}$
> >
> > We appreciate your comment. As the reviewer pointed out, our expression “per-tensor quantization schemes facilitate higher parallelism for implementation compared to per-channel quantization schemes (Nagel et al., 2021)” could be potentially unclear and/or confusing. Our intention was to emphasize that per-tensor quantization can take advantage of integer matrix-to-matrix multiplication API/function calls (e.g., integer forms of cuBLAS and cuDNN that utilize tensor cores) in commercial GPUs (while per-channel can partition a matrix-to-matrix form into multiple vector-to-vector forms with a lower degree of parallelism). Reflecting the reviewer's comment, we will remove the reference in the sentence and revise the sentence as follows: “Per-tensor quantization schemes might enable integer matrix-to-matrix multiplication API/function (Migacz, 2017) that can facilitate efficient inference of quantized models.”
> >
> > Szymon Migacz. 8-bit inference with TensorRT. In NVIDIA GPU Technology conference, 2017.
> >
> > ------------------------------------------------------------------------

---

> > > ### Comment · Reviewer_oCmz · 2022-12-02
> > > **Thanks for the detailed response**
> > >
> > > Thanks for the detailed response and the provided extra experiments and insights. I adjusted my score accordingly.
> > >
> > > Here some further suggestions based on the changes in the revised manuscript:
> > > * The insight on $s_1$ is very interesting (Q6). Based on that, it actually seems to me that the benefit of FlexRound does not only come from the element-wise division, but is rather two fold: 1) jointly learning the rounding and scale (which is not possible with AdaRound) and 2) the element-wise division. I would suggest to refer to these important results and insights from the main text (or even move the study there if space permits).
> > > * I would suggest to make table 1, 2 and 3 using the same models (maybe BRECQ?), otherwise it is confusing for a reader to since rows that should be the same are not. Then the appendix can have the results with the other pre-trained models as you already have in appendix C.

---

> > > > ### Author Response · Authors · 2022-12-05
> > > > **Dear Reviewer oCmz [3]**
> > > >
> > > > Dear Reviewer oCmz,
> > > >
> > > > We sincerely appreciate your helpful and detailed comment.
> > > >
> > > > As suggested, in the final revision, we will add the importance of learning $s_1$ in the main text and modify Table 1 by using pre-trained models available from the BRECQ github repository.
> > > >
> > > > Once again, thank you for your time and efforts in reviewing our paper.

---

### Author Response · Authors · 2022-11-09
**General Response to the Clarity of BRECQ and QDrop Experimental Results**

We would like to appreciate all reviewers’ constructive and helpful feedback.

Since all reviewers present a concern about experimental results (specifically model accuracy) on BRECQ and QDrop, we would like to separately elaborate how we obtained mismatched accuracy numbers in Table 2 and 3 compared to the original manuscripts (of the authors of BRECQ and QDrop) as follows:

(1) In our initial manuscript, we used pre-trained models available in the official PyTorch repository whereas the authors of BRECQ and QDrop have designed their own pre-trained models (that significantly outperform those from the official PyTorch repository). We agree that it might be confusing if we choose different pre-trained models different from the ones that the manuscripts of BRECQ and QDrop selected. Accordingly, in the revised manuscript, we updated Table 2 and Table 3 to address reviewers concerns. We believe that our claims and discussions are still valid even after updating two tables.

(2) We adopt $\textit{per-tensor}$ linear $\textit{symmetric}$ weight quantization while the authors of BRECQ and QDrop adopt $\textit{per-channel}$ linear $\textit{asymmetric}$ weight quantization in their manuscripts.

The following tables (of experiments based on pre-trained models available from the BRECQ github repository) are included in our revised manuscript.

(When activations are not quantized)
| Method | \# Bits (W./A.) | ResNet-18 | ResNet-50 | MobileNetV2 |
|:---------|:-----------------:|:--------------:|:--------------:|:----------------:|
| Full-precision | 32 / 32 | 71.00 / 89.97 | 76.63 / 93.04 | 72.62 / 90.67 |
| B + AdaQuant | 4 / 32 | 67.50 / 87.75 | 72.79 / 90.77 | 15.17 / 32.89 |
| B + AdaRound | 4 / 32 | 70.18 / 89.38 | 75.86 / 92.62 | 69.46 / 88.85 |
| B + FlexRound (Ours) | 4 / 32 | $\mathbf{70.28} / \mathbf{89.44}$ | $\mathbf{75.95} / \mathbf{92.68}$ | $\mathbf{70.82} / \mathbf{89.67}$ |
| B + AdaQuant | 3 / 32 | 57.09 / 80.82 | 52.13 / 75.22 | 0.20 / 0.79 |
| B + AdaRound | 3 / 32 | $\mathbf{68.79} / \mathbf{88.62}$ | $74.31 / 91.81$ | $62.51 / 84.52$ |
| B + FlexRound (Ours) | 3 / 32 | 68.65 / 88.54 | $\mathbf{74.38} / \mathbf{91.81}$ | $\mathbf{66.87} / \mathbf{87.56}$ |
| B + AdaQuant | 2 / 32 | 0.23 / 0.92 | 0.10 / 0.50 | 0.10 / 0.50 |
| B + AdaRound | 2 / 32 | 61.99 / 84.81 | 48.47 / 77.09 | 39.57 / 66.18 |
| B + FlexRound (Ours) | 2 / 32 | $\mathbf{62.57} / \mathbf{84.84}$ | $\mathbf{63.67} / \mathbf{85.72}$ | $\mathbf{46.04} / \mathbf{72.48}$ |

(When activations are also quantized)
| Method | \# Bits (W./A.) | ResNet-18 | ResNet-50 | MobileNetV2 |
|:---------|:-----------------:|:--------------:|:--------------:|:----------------:|
| Full-precision | 32 / 32 | 71.00 / 89.97 | 76.63 / 93.04 | 72.62 / 90.67 |
| B + AdaRound | 4 / 4 | 69.18 / 88.85 | 74.44 / 91.80 | 61.05 / 83.30 |
| B + FlexRound (Ours) | 4 / 4 | $\mathbf{69.32} / \mathbf{88.83}$ | 74.56 / 91.87 | 63.74 / 85.01 |
| Q + AdaRound | 4 / 4 | 69.20 / 88.96 | 74.90 / 92.15 | 65.42 / 86.23 |
| Q + FlexRound (Ours) | 4 / 4 | 69.26 / 88.81 | $\mathbf{75.08} / \mathbf{92.20}$ | $\mathbf{66.66} / \mathbf{87.21}$ |
| B + AdaRound | 3 / 3 | 64.83 / 86.12 | 67.01 / 87.28 | 3.74 / 11.54 |
| B + FlexRound (Ours) | 3 / 3 | 64.99 / 85.93 | 68.29 / 87.89 | 25.43 / 48.28 |
| Q + AdaRound | 3 / 3 | $\mathbf{65.71} / \mathbf{86.96}$ | 70.49 / 89.93 | 39.86 / 66.00 |
| Q + FlexRound (Ours) | 3 / 3 | 65.43 / 86.60 | $\mathbf{70.74} / \mathbf{89.78}$ | $\mathbf{51.49} / \mathbf{76.90}$ |

When using pre-trained models available from the BRECQ github repository, we could attain the results more close to those in the original BRECQ and QDrop papers than before. Please find our revised manuscript.

---

### Author Response · Authors · 2022-11-12
**General Response for Additional PTQ results on Finetuned models in NLG tasks**

Dear reviewers.

As for the evaluation of quantized pre-trained language models, the performance (i.e., accuracy) of quantized OPT (by Q+AdaRound or Q+FlexRound) is not close to that of full-precision OPT, while GPT-Neo can be quantized without noticeable accuracy degradation. To investigate whether such an observation is also valid for finetuned OPT or not, we conduct additional experiments on finetuned OPT and GPT-Neo with Wikitext2 and PTB dataset. As shown in the table below, quantized model’s performance of finetuned OPT turns out to be close to full-precision performance. Experimental results and detailed settings are attached in the appendix.
| Dataset   | Method             | GPT-Neo 125M      | GPT-Neo 1.3B      | GPT-Neo 2.7B      | OPT 125M          | OPT 1.3B          | OPT 2.7B          |
|:---------|:------------------|:-----------------:|:-----------------:|:-----------------:|:-----------------:|:-----------------:|:-----------------:|
| WikiText2 | Full-precision     | $21.96$           | $12.09$           | $10.78$           | $19.85$           | $11.52$           | $10.27$           |
| WikiText2 | Q+AdaRound         | $30.52$           | $12.47$           | $14.09$           | $27.96$           | $12.66$           | $10.97$           |
| WikiText2 | Q+FlexRound (Ours) | $\\mathbf{24.30}$ | $\\mathbf{12.37}$ | $\\mathbf{12.43}$ | $\\mathbf{21.43}$ | $\\mathbf{12.02}$ | $\\mathbf{10.63}$ |
| PTB       | Full-precision     | $24.20$           | $16.09$           | $14.70$           | $16.50$           | $11.62$           | $10.80$           |
| PTB       | Q+AdaRound         | $31.40$           | $16.63$           | $19.80$           | $20.28$           | $13.00$           | $12.02$           |
| PTB       | Q+FlexRound (Ours) | $\\mathbf{26.03}$ | $\\mathbf{16.32}$ | $\\mathbf{16.87}$ | $\\mathbf{17.68}$ | $\\mathbf{12.22}$ | $\\mathbf{11.29}$ |

---

### Author Response · Authors · 2022-11-14
**Revision summary**

We appreciate all reviewers for their constructive feedback. Based on their comments, we revised the paper by making the following changes. The modified part is highlighted in blue in the revision.

Major updates:
- We have replaced Table 2 and 3 with the result based on pre-trained models from the BRECQ github repository. (Reviewer oCmz, mQzD, y3vc)
- We have modified our manuscript to express “we are the first to conduct extensive experiments in the form of per-tensor uniform PTQ reconstruction on natural language generation as well as image classification and natural language understanding” in the appendix and Section 1. (Reviewer oCmz)
- We have clarified that only AdaQuant may change the sign of a weight in Section 3.2. (Reviewer oCmz)
- We have removed the expression that AdaRound requires numerous hyper-parameters in Section 2. (Reviewer oCmz)
- We have added the ablation study on FlexRound with $s_1$ fixed in the appendix. (Reviewer oCmz)
- We have added the result of combining an additive approach with a division-based approach in the appendix. (Reviewer oCmz, vGda)
- We have replaced [⋅] with ⌊⋅⌉. (Reviewer oCmz)
- We have modified  the sentence about high parallelism in Section 3.2. (Reviewer oCmz)
- We have added the reason why all entries of $\mathbf{S}_2$, $\mathbf{s}_3$, and $\mathbf{s}_4$ are initialized to be ones in Section 3.2. (Reviewer mQzD)
- We have added the derivation of ${{\partial\mathcal{L}} \over {\partial \mathbf{S'}}}$ in the appendix. (Reviewer mQzD)
- We have added some recent papers on PTQ in Section 2. (Reviewer mQzD)
- We have added the ablation study on how much data is needed to learn $\mathbf{S}$ in the appendix. (Reviewer vGda)
- We have moved Section 3.3 to Section 4.1. (Reviewer y3vc)
- We have added additional PTQ results on fine-tuned models in NLG tasks in the appendix.

We believe that our paper becomes much stronger and clearer with this revision, thanks to the reviewers for constructive suggestions.

---

### Author Response · Authors · 2022-11-16
**Dear Reviewers**

Dear Reviewers,

We sincerely thank you for your time and efforts in reviewing our paper. We gently remind you that we have responded to your insightful and constructive comments and faithfully reflected them in the revised manuscript. We would appreciate it if you check our responses and the revision.

Thanks, Authors

---

### Decision · Program_Chairs · 2023-01-20

**Decision:**

Reject

**Justification For Why Not Higher Score:**

It is not clear or convincing why the current design of the method can improve quantization performance. Some concerns remain about the reproducibility of existing methods in experiments.

**Justification For Why Not Lower Score:**

N/A

**Metareview: Summary, Strengths And Weaknesses:**

In this paper, the authors proposed a new post-training quantization. Though in the rebuttal, the authors provided explanations and additional experiments, some points are still not clear to reviewers. For instance, the key reason for the proposed method that improves quantization performance over existing methods or simple baselines is not clear or convincing. There are concerns about the reproducibility of existing methods in experiments.

Overall, it is a potential promising work, but needs to be further polished for publication.